# Annual and semiannual cycles of midlatitude near-surface temperature and tropospheric baroclinicity: reanalysis data and AOGCMs simulations

Valerio Lembo[1], Isabella Bordi[2,3], Antonio Speranza[2]

[1]Department of Biological and Environmental Sciences and Technologies (DiSTeBA), University of Salento, Lecce, Italy
[2]National Interuniversity Consortium for the Physics of the Atmosphere and Hydrosphere (CINFAI), Rome, Italy
[3]Department of Physics, Sapienza University of Rome, Rome, Italy

*Correspondence to*: Isabella Bordi (isabella.bordi@cinfai.it), Valerio Lembo (valerio.lembo@unisalento.it)

**Abstract.** Seasonal variability of near-surface air temperature and baroclinicity from the ECMWF ERA-Interim (ERAI) reanalysis and six coupled atmosphere-ocean general circulation models (AOGCMs) participating in the Coupled Model Intercomparison Project phase 3 and 5 (CMIP3 and CMIP5) are examined. In particular, the annual and semiannual cycles of hemispherically averaged fields are studied using spectral analysis. The aim is to assess the ability of coupled general circulation models to properly reproduce the observed amplitude and phase of these cycles, and investigate the relationship between near-surface temperature and baroclinicity (coherency and relative phase) in such frequency bands. The overall results of power spectra agree in displaying a statistically significant peak at the annual frequency in the zonally averaged fields of both hemispheres. The semiannual peak, instead, shows less power and in the NH seems to have a more regional character, as is observed in the North Pacific Ocean region. Results of bivariate analysis for such a region and Southern Hemisphere midlatitudes show some discrepancies between ERAI and model data, as well as among models, especially for the semiannual frequency. Specifically: (i) the coherency at the annual and semiannual frequency observed in the reanalysis data is well represented by models in both hemispheres; (ii) at the annual frequency, estimates of the relative phase between near-surface temperature and baroclinicity are bounded between about $\pm 15^{\circ}$ around an average value of $220^{\circ}$ (i.e., approximately 1 month phase shift), while at the semiannual frequency model phases show a wider dispersion in both hemispheres with larger errors in the estimates, denoting increased uncertainty and some disagreement among models. The most recent CMIP climate models (CMIP5) show several improvements when compared with CMIP3 but a degree of discrepancy still persists though masked by the large errors characterizing the semiannual frequency. These findings contribute to better characterize the cyclic response of current global atmosphere-ocean models to the external (solar) forcing that is of interest for seasonal forecasts.

# 1 Introduction

The seasonal cycle of the heating of the atmosphere is one of the most prominent features of the Earth's climate (e.g., Kiehl and Trenberth, 1997; Trenberth and Stepaniak, 2004). ). A recent study by Donohoe and Battisti (2013) suggested that while in the annual average heating is dominated by upward energy fluxes from the surface, such as longwave, latent and sensible heat fluxes (e.g. Wild et al., 2013), most of the seasonal heating (i.e., the heating variability after subtracting the annual mean) is attributable to the direct shortwave absorption within the atmosphere, with an amplitude that is quite constant throughout the troposphere. It is estimated that the variation of solar insolation, due to the orbital movement of the Earth around the Sun, accounts for about 90 % of the total variance of surface temperature (Trenberth, 1983).

Observations show that at midlatitudes the annual harmonic is by far the largest component of the seasonal cycle, while other sub-harmonics capture only the finer structure of the cyclic variation. Approaching the equatorial regions, the seasonal cycle has a more complicated behavior and the annual and semiannual harmonics are both large components. The semiannual signal characterizes also high-latitude regions. Furthermore, there are observed differences in the seasonality of atmospheric temperature and eddy activity between the Northern and Southern Hemispheres (hereafter NH and SH), with the NH exhibiting stronger seasonal variation due to the larger portion of land surface (Peixoto and Oort, 1992).

It is worth noticing that the annual cycle of the global mean net radiation at the top of the atmosphere (defined as the difference between the downward absorbed solar radiation and the outgoing longwave radiation) is of the order of 20 W/m$^2$ or, integrated globally, about 10 PW (e.g., Fasullo and Trenberth 2008). On the other hand, the global climate forcing (i.e., the change of the planetary energy balance) due to the total greenhouse gases is estimated to be of less magnitude, about 3 W/m$^2$ or 1.5 PW, based on the change in gas concentrations since 1750 (NOAA 2016). Thus, the annual/semiannual cycles are the leading natural changes which the atmosphere experiences every year, and their proper representation can be considered the starting point for any climate projection. This applies also in relation to GCM climate simulations that, for example, have highlighted changes in the seasonality of surface temperature (amplitude and phase) in response to the increasing greenhouse gases concentration (e.g., Dwyer et al. 2012 and references therein).

Since the first efforts in understanding the climate impact of increasing atmospheric carbon dioxide (Hansen et al., 1981), surface temperature is taken as a proxy of globally averaged temperature in climate change studies. Due to its decorrelation length (about 1200 km; Hansen and Lebedeff, 1987) and its key role in radiative transfer, it is considered a useful variable for studying large-scale processes and climate sensitivity. Large-scale seasonal variability of surface temperature has been successfully simulated in earlier energy balance models (EBMs), with "passive" ocean and atmospheric heat transports parameterized as a diffusive process (e.g., North et al., 1983; Kim and North, 1992). The simulation of the seasonal cycle of near-surface air temperature has long been considered a test of performance of climate models (Covey et al., 2000). The ability of Atmosphere-Ocean General Circulation Models (AOGCMs) to simulate a reasonable seasonal cycle is, in fact, a necessary condition for confidence in their prediction of long-term climatic behavior, including global changes (e.g., Bye et al., 2013). Furthermore, the amplitude of the seasonal cycle in surface temperature has been used to verify the climate

sensitivity of models (Lindzen et al., 1995; Knutti et al., 2006), and the magnitude of the seasonal cycle has been found to be a good predictor of the magnitude of decadal variability in regional surface temperatures (Huybers and Curry, 2006).

The seasonal cycle is also the main modulator of weather variability at midlatitudes. It is well known that transient eddies, which transport heat and momentum in the extratropics, are generated by the horizontal temperature gradient in the mean flow through a process of energy conversion from available potential to kinetic. Such a temperature gradient (often referred to as baroclinicity) is the basis for eddies' development in linear instability theories, nonlinear models of geostrophic turbulence, and observations (e.g., Lindzen and Farrell, 1980; Held and Larichev, 1996; Hoskins and Valdes, 1990). Several analyses have pointed out that there are regions of enhanced eddy activity (i.e., storm tracks), where weather systems preferentially grow through baroclinic instability and subsequently decay. In the NH, such regions lie downstream, and slightly poleward, of the cores of the jet stream over the North Atlantic and Pacific Oceans (e.g., Blackmon et al., 1977). It has also been shown that, like most of atmospheric processes, baroclinic activity is characterized by a seasonal cycle. Previous studies provided evidences for some kind of midwinter suppression of baroclinic wave activity in the NH Pacific, with large variances during spring and autumn (see Nakamura 1992 and references therein). This finding was supported by the harmonic analysis of air temperature and sea level pressure fields carried out by Yashyayaev and Zveryaev (2001), by observations as illustrated by Chang (2003), or the recent study by Chen et al. (2012).

In SH midlatitudes, an important signature of the seasonal cycle is the Southern Hemisphere Semiannual Oscillation (SAO), a coupled ocean-atmosphere phenomenon that involves the different annual cycles of temperature between the Antartic polar continent and the surroundings midlatitude oceans (van Loon, 1967; Meehl, 1991). The twice-yearly intensification of temperature gradient between midlatitudes (ocean dominated) and polar latitudes (continental) is associated with a fluctuation in the storm activity.

The persistent character of these regional features in both hemispheres suggests that some processes are acting on the system to continuously sustain the storm tracks, i.e. external forcing and/or feedback mechanisms. Following Lorenz (1979), variations of weather and climate are forced or free according to whether they result from changes in the external conditions or not. Such a characterization is usually analyzed by examining the correlation between two variables of interest, for example the eddy heat fluxes and baroclinicity (e.g., Stone et al., 1982). Thus, investigating the possible relationship between baroclinicity and surface temperature seasonal cycles may help to better understand the role played by the latter (which is directly related to solar forcing and heat fluxes) in modulating the atmospheric processes at midlatitudes.

In the present paper, the annual and semiannual harmonics of midlatitude surface temperature and baroclinicity in both hemispheres are studied by applying spectral analysis to ERA-Interim reanalysis and Coupled Model Intercomparison Project (CMIP) data. Furthermore, the relationship (coherency and relative phase) between surface temperature and baroclinicity, which is here estimated using the maximum Eady growth rate, is investigated. The aim is to assess the ability of AOGCMs to properly reproduce the amplitude and phase of the observed annual and semiannual periodicities that are of interest for seasonal forecasts.

The structure of the paper is as follows. Section 2 describes data and methods, while Section 3 provides a description of the main results obtained from spectral analysis. A summary and conclusions are given in the final section.

## 2 Data and methods

### 2.1 Data

Data used for the study are derived from (i) the ERA-Interim archive and (ii) the Coupled Model Intercomparison Project phase 3 (CMIP3) multi-model dataset, including the most recent phase 5 of the project (CMIP5) for comparison. ERA-Interim (hereafter ERAI) is the latest global atmospheric reanalysis product delivered by the European Centre for Medium Range Weather Forecasts (ECMWF; Berrisford et al., 2011). The data assimilation system used to produce ERAI is based on the 2006 release of the Integrated Forecasting System (IFS) that includes a 4-dimensional variational analysis (4D-Var) with a 12-hour analysis cycle. In each cycle, all available observations (in situ measurements, radiosondes, satellites, etc.) are combined with prior information from the forecast model to estimate the evolving state of the atmosphere (Dee et al., 2011). The model has T255 horizontal spectral resolution (~ 80 km) with 60 isobaric levels from the surface up to 0.1 hPa. ERAI improved on several deficiencies reported in the previous reanalysis ERA-40, in particular the water cycle that was too wet in the tropics and breaks in the time series of some products that are likely related to the introduction of satellites into the assimilation scheme (Poli et al., 2010).

In the present study, monthly means of daily means of 2-meter temperature (T2m) and tropospheric temperature ($T$) covering the period January 1979–September 2015, at 1-degree horizontal resolution and 8 vertical levels in the troposphere (from 1000 hPa to 300 hPa), are considered and analyzed.

Under the World Climate Research Programme (WCRP), the Working Group on Coupled Modelling (WGCM) established the CMIP as a standard experimental protocol for studying the outputs of coupled AOGCMs. The phase 3 multi-model dataset is here used: it accounts for historical climate reconstruction performed by models, from the pre-industrial era to the beginning of the 21st century (Meehl et al., 2007). The integration period, the radiative forcing parameterization and the horizontal resolution vary from model to model. Whatever is the model vertical resolution, a minimum number of pressure levels for vertical discretization of 3D atmospheric outputs is required to be 17 specified levels from 1000 to 10 hPa (see for details http://www-pcmdi.llnl.gov/ipcc/model_documentation/ipcc_model_documentation.php).

Based on the intraseasonal and interannual variability analysis at milatitudes carried out by Lucarini et al. (2007), for the purpose of the present study, a subset of 6 models over 27 were chosen (see Table 1). The selection has been made taking into account models' performance, when compared with reanalyses, and the horizontal resolution. For CMIP3, the subset comprises one high-resolution model (MIROC3.2), four medium-resolution models (CGCM3.1, ECHAM5/MPI-OM, FGOALS-g1.0, GFDL-CM2.1) and one coarse resolution model (INM-CM3.0). According to Lucarini et al. (2007), while CGCM3.1, GFDL-CM2.1 and MIROC3.2 high-resolution compare well with the reanalysis, ECHAM5/MPI-OM and

FGOALS-g1.0 have been found to overestimate both intraseasonal and interannual variability. INM-CM3.0 is here considered as an example of a coarse-resolution model.

Aiming to verify any improvement in the model representation of the features of interest, the most recent multi-model dataset CMIP5 has been also considered (Taylor et al., 2012; http://cmip-pcmdi.llnl.gov/cmip5/guide_to_cmip5.html). CMIP5 differs from earlier phases in the wider variety of scientific issues to be addressed, the larger number of models participating into the project, the generally higher spatial model resolution, a richer set of output fields archived, and the two time scales of the experiments (i.e., long-term and near-term decadal prediction runs). For the present analysis, 6 CMIP5 models, which represent the updated versions of the ones described above, have been selected (see Table 1). It is worth noticing that, in the new ensemble dataset (CMIP5), the model ECHAM5/MPI-OM (CMIP3) has been replaced by MPI-ESM-MR, which is the ECHAM6 model coupled with the same ocean model MPIOM used in phase 3. Furthermore, INM-CM4 and MIROC5 horizontal resolutions are close to those of the medium-resolution CMIP3 models. As for CMIP3, the same 17 pressure levels are required for the atmospheric outputs and 8 isobaric levels (up to 300 hPa) have been considered for the representation of the tropospheric mean conditions.

## 2.2 Methods

*Baroclinicity*

Synoptic eddy activity in midlatitudes has long been related to the baroclinic instability that converts available potential energy of the time mean flow to eddy kinetic energy (e.g., Charney, 1947; Eady, 1949; Lorenz, 1955). One of the measures of atmospheric baroclinicity is the maximum growth rate of the linear Eady model, which has been shown to be a useful estimate of the growth rate of the most rapidly growing instability in a wide range of baroclinic instability problems (Lindzen and Farrell, 1980). The maximum Eady growth rate has been found to be a suitable parameter to quantify the geographical location and intensity of the storm tracks (i.e., Hoskins and Valdes, 1990; Wu et al., 2011), to evaluate the impact of increasing greenhouse gases and sulphate aerosols on extratropical cyclone activity (Carnell and Senior, 1998; Geng and Sugi, 2003), and to measure the baroclinicity of the mean state of the atmosphere (e.g., Heo et al., 2012). In the present paper, it is applied to the reanalysis and model data. Making use of the thermal wind equation, an accurate estimate of the dimensional growth rate (hereafter referred to as baroclinicity index) is given by (Lindzen and Farrell, 1980):

$$\sigma_{BI} = 0.3125 \frac{f}{N} \left| \frac{\partial U}{\partial z} \right| \approx -0.3125 \frac{g}{a \bar{T} N} \left| \frac{\partial \bar{T}}{\partial \phi} \right| \tag{1}$$

with $f$ the Coriolis parameter, $U$ the zonal wind component, $a$ the Earth's radius, $\phi$ the latitude, $g$ the gravity acceleration, $\bar{T}$ the mean temperature in the troposphere and $N$ the Brunt-Väisälä frequency, a measure of the static stability. As can be noted, the dry Eady growth rate (moist processes generally lead to an increase of the rate; see Emanuel et al., 1987) is

affected by changes of both meridional temperature gradient (or vertical wind shear) and static stability. However, in the present study, $N$ is taken to be constant and equal to the mean tropospheric value of $1.2 \cdot 10^{-2}$ s$^{-1}$ (Holton, 2004). The average over both hemispheres here considered, in fact, cuts off most of the variability of the Brunt-Väisälä frequency that occurs mainly near the surface and at high latitudes (Simmons and Hoskins, 1980).

As introduced in the previous section, surface temperature is taken as the air temperature at 2 meters height, while tropospheric temperature is averaged over the 8 pressure levels between 1000 and 300 hPa for both ERAI and AOGCMs. Both fields, surface temperature and tropospheric temperature, are separately considered for the NH and the SH. In the NH, baroclinicity index is computed over the latitude band 30$^{o}$–60$^{o}$N, the area of most intense baroclinic activity (Dell'Aquila et al., 2007a); in the SH, the latitude band 30$^{o}$–70$^{o}$S is considered where the bulk of baroclinic and low-frequency planetary

waves activity is observed (Dell'Aquila et al., 2007b). The meridional temperature gradient in (1) is computed as the difference between the two boundaries of the considered midlatitude belt, while vertical average of tropospheric temperature is obtained without weighting for the air mass. Furthermore, we notice that the baroclinicity index here considered, which is proportional to the tropospheric meridional temperature gradient, is independent from the zonal mean T2m; it would be not so if the meridional gradient of T2m would be considered in place of T2m, due to the expected degree of dependence

between tropospheric and near-surface temperature gradients.

*Spectral analysis*

The spectral components of surface air temperature and baroclinicity have been estimated and tested for statistical significance by using the Multi Taper Method (MTM), which is a nonparametric technique widely applied to problems in the

analysis of geophysical signals (Thomson, 1982; Percival and Walden, 1993). MTM attempts to reduce the variance of spectral estimates by using a small set of tapers (or spectral windows) rather than single data taper used by other methods like Blackman-Tukey. Data are pre-multiplied by orthogonal tapers (or eigentapers), which are constructed to minimize the spectral leakage due to the finite length of the data series, and, by performing a Fourier Transform, a set of independent power spectral density (PSD) estimates is computed. The optimal tapers belong to a family of functions known as Discrete

Prolate Spheroidal Sequences (DPSS) or Slepian sequences, which solve the variational problem of minimizing leakage outside of a frequency band of half bandwidth $pf_n$, where $f_n = 1/(N\,\Delta t)$ is the Rayleigh frequency (i.e., the minimum frequency that can be resolved by a finite duration time window), $\Delta t$ the sampling time, and $p$ an integer. Typically, the number of tapers used $K$ should be less than $(2p-1)$, i.e. the minimum number of tapers that provide small spectral leakage (Ghil et al., 2002). Thus, the bandwidth $2pf_n$ and the number of tapers $K$ represent the key parameters for the stability of the

power spectral estimate, which become:

$$S = \frac{1}{K}\sum_{k=1}^{K}\Delta t \left| \sum_{t=1}^{N} DPSS_{t,k}\, x(t)\, e^{-i2\pi f \Delta t} \right|^2 \qquad\qquad (2)$$

where $x(t)$ denotes the signal, $\mathrm{DPSS}_{t,k}$ is the $k^{\text{th}}$ taper function at time point $t$.

The confidence intervals in the PSD estimates are computed using the chi-squared approach at the 97.5 % confidence level.

To investigate the relation between the two fields of interest (surface temperature and baroclinicity) in the frequency domain, the bivariate spectral analysis is performed. The relationship between two time series in the Fourier domain can be expressed in terms of the cross-spectrum, the phase difference, and the cross-coherence, which is defined as follows.

Being $P_A(f)$ and $P_B(f)$ be the complex Fourier spectra of two time series, the cross-spectrum is defined as:

$$P_{AB} = P_A\left(f\right) \cdot P_B^*\left(f\right) \tag{3}$$

with the asterisk denoting the conjugate. The phase difference can be written as:

$$\Phi_{AB}\left(f\right) = \arg\left\langle P_{AB}\left(f\right)\right\rangle \tag{4}$$

with the convention that a value between $0^{\text{o}}$ and $180^{\text{o}}$ means that $A$ leads $B$ (in our case T2m leads $\sigma_{\text{BI}}$) and vice versa for a value between $180^{\text{o}}$ and $360^{\text{o}}$.

The coherence spectrum is a measure of the correlation of the two spectra as a function of frequency and can be written as:

$$COH_{AB}\left(f\right) = \frac{\left|\left\langle P_{AB}\left(f\right)\right\rangle\right|}{\sqrt{\left\langle \left|P_A\left(f\right)\right|^2 \right\rangle \left\langle \left|P_B\left(f\right)\right|^2 \right\rangle}} \tag{5}$$

The angle brackets denote the expectation value and can be approximated by a mean over many short spectra (Von Storch and Zwiers, 1999).

## 3 Results

To provide a frame of reference for the subsequent analysis, we first present the time series of midlatitude surface temperature and baroclinicity in both hemispheres as derived by ERAI reanalysis. The seasonal behavior of some averaged fields (i.e., TOA solar radiation, mean sea level pressure, and meridional gradient of geopotential height field at 300 hPa and 500 hPa) introduce and complement this preliminary analysis.

For illustrative purposes, the spectral analysis for selected CMIP3 coupled models follows; results for both hemispheres are illustrated and compared with those from the most recent CMIP5 models.

**3.1 Time series analysis**

In Figure 1 the observed seasonal variability in both hemispheres of ERAI near-surface temperature and baroclinicity are displayed with that of solar radiation at TOA, the meridional gradient (defined as the difference between 30° and 70°) of the geopotential height (GPH) field at 300 hPa and 500 hPa, and the mean sea level pressure (MSLP) variance in the midlatitude belt. The annual mean cycles have been computed by averaging daily values over the entire time record of the reanalysis data and applying 15-day moving average to filter out the high-frequency variability; then, they have been standardized for comparison. For illustrative purposes, the baroclinicity, GPH, and MSLP variance cycles have been reversed so that their maxima occur in summer as for the other fields.

From Figure 1a (NH) it can be noted that, moving from January to December, the incoming solar radiation precedes the other signals: in particular, it is followed by surface temperature and baroclinicity index. The meridional geopotential gradient at 300 hPa (not shown) and 500 hPa appears in phase with the baroclinicity index, while surface temperature shows a delay of 30–45 days with respect to the incoming solar radiation, and of about 20 days with respect to baroclinicity and geopotential gradients. The annual cycles for the SH (Figure 1b) show similar features with a more pronounced semiannual oscillation in the geopotential gradient and baroclinicity index. A hint of SAO is also observed in MSLP variance cycle. As documented by Meehl (1991), in fact, the SAO phenomenon is evident in monthly mean maps of observed MSLP: the trough of MSLP minimum is farthest south and deepest in March and September, while it is farthest north and weakest in June and December (their Figure 1). Such a movement of the circumpolar trough is associated with changes in the cyclone activity in extensive areas and evidences throughout the depth of the troposphere (for example in 500 mb temperature). Furthermore, the standardization allows estimating the relative amplitude of the annual cycles that can be grouped as: on one hand, surface temperature and MSLP variance, and on the other hand, GPH meridional gradients and baroclinicity index. It is worth noticing that results are in agreement with those obtained by Donohoe and Battisti (2013) showing that atmospheric temperature lags the insolation by approximately 30 days in the northern and 40 days in the southern extratropics, respectively.

Monthly mean time series (1979–2015) of both ERAI surface temperature and baroclinicity averaged over the NH and the SH midlatitude belts are shown in Figure 2 and 3, respectively. Figures clearly show that the periodic components account for most of the variability of the time series: the annual cycle characterizes the time series with larger amplitude in the NH than in the SH, likely explained by the different land distribution in the two hemispheres and by the difference in the heat capacity of land and ocean. As known, in both hemispheres surface temperature follows solar heating and is dominated by the annual cycle, with a weak semiannual component (Peixoto and Oort, 1992). Moreover, the amplitude of the NH surface temperature annual cycle is roughly twice as strong as the SH and they are in phase opposition so that, on the global scale, the mean surface temperature has a weaker residual seasonal cycle (Covey et al., 2000). Also, to be noted is the phase opposition between surface temperature and baroclinicity index, with the latter showing a semiannual modulation of the

annual cycle (Figure 3 when compared with Figure 2). Such a modulation, which in the SH is a signature of SAO phenomenon (van Loon, 1967; Meehl, 1991), is found also in the NH.

For a comparison, the time series of surface temperature and baroclinicity from the 6 selected CMIP3 models are shown in Figure 4 for the common time section 1979–1999. For the sake of clarity, the time series of T2m and $\sigma_{BI}$ are standardized

and plotted into the same graphs. As expected, there is an opposite phase relationship between baroclinicity and surface temperature; furthermore, as shown by the reanalysis data, there is evidence in model data of the semiannual oscillation in both hemispheres, suggesting the robustness of such a feature in the time series. A few outliers are found in baroclinicity records, likely related to anomalous values in air temperature data.

## 3.2 Spectral estimates

The power spectral density (PSD) estimates of ERAI mean baroclinicity and surface temperature, for the NH and SH midlatitudes, computed using the MTM method are shown in Figure 5 (p = 3, K = 5). Peaks are tested for significance at 95 % level relative to the null hypothesis of a red noise background estimated from the data.

As already suggested by Figure 2 and 3, in both surface temperature and baroclinicity spectra most power is at the annual frequency, leading to a peak that is statistically significant. A secondary harmonic lies at the semiannual frequency, which,

for the baroclinicity index, is statistically significant or marginally significant in both hemispheres (i.e., the red noise spectrum lies within the confidence interval of PSD estimate), while for surface temperature it is not in the NH and marginally statistically significant in the SH. This means that in NH midlatitudes surface temperature is dominated by the annual cycle modulated by a weaker semiannual periodicity embedded into the red noise background, suggesting a more regional nature of the semiannual component. Since we are interested to investigate the relationship between surface

temperature and baroclinicity annual/semiannual cycles, we selected ocean regions in NH midlatitudes where baroclinic eddies are known to be particularly active and requiring that also the semiannual component in T2m is present (statistically significant). Candidate regions of interest are: NH oceans, Pacific Ocean, and Atlantic Ocean, all in the latitude band 30–60$^o$N. For these regions, power spectra of baroclinicity index (not shown) are characterized by statistically significant annual and semiannual cycles, with the exception of the Atlantic Ocean where no semiannual harmonic is found. The power spectra

estimates of near-surface temperature are shown in Figure 6: as can be noted, for the Pacific Ocean the red noise spectrum falls within the confidence interval of the PSD estimate. Hence, for the NH, we decided to focus the subsequent analysis on this region.

Before investigating whether there is any (amplitude or phase) direct relationship between zonal mean near-surface temperature and baroclinicity we have computed the absolute phases of observed (ERAI) and modelled (CMIP3 and CMIP5)

$\sigma_{BI}$, and T2m (Table 2). Almost all phases are lagged less than about 1 month with respect to ERAI (i.e., about 30$^o$ for the annual phase and 60$^o$ for the semiannual one) and better agreement is found for CMIP5. Being 1 month the sampling time of the time series, results suggest that models are to a reasonable extent able to reproduce the phases of the reanalysis data. However, the observed coherences and relative phases between $\sigma_{BI}$ and T2m may be partially affected by such not a perfect

in phase relationship between model data and reanalysis. Thus, this aspect should be taken into account as a possible source of uncertainty when results of bivariate phase spectra analysis are compared and interpreted.

Results of bivariate spectral analysis (coherency and phase spectra) are shown in Figure 7 for the reanalysis ERAI and in Figures 8–9 for model outputs, respectively. According to the adopted convention, in case of a phase value between $0^o$ and $180^o$ surface temperature is leading with respect to baroclinicity, and vice versa in case of a phase value between $180^o$ and $360^o$.

Consistently with Figure 5 and 6, the coherency spectra between observed surface temperature and baroclinicity in the NH and the SH (Figures 7a and 7c) show peaks at the annual frequency very close to the level of 1 (total coherency), implying strong correlation between the two time series. Similarly, at the semiannual frequency high coherency is found in both hemispheres. This finding provides evidence for the already mentioned SAO and, more importantly, confirms the relationship between the two variables at the semiannual periodicity also in the NH Pacific Ocean.

For most frequencies, phase spectra (Figures 7b and 7d) depart from the condition of null phase. In particular, in the NH Pacific Ocean at the annual frequency, a phase shift of about $213^o$ is found (i.e., about $30^o$ with respect to the opposition of phase), and about $258^o$ at the semiannual one. In the SH, at the annual frequency a phase shift of about $214^o$ is observed and at the semiannual frequency about $235^o$, showing, in both cases, a delay of the mean surface temperature with respect to the baroclinicity index.

Results obtained for the annual frequency in the NH Pacific Ocean and the SH are consistent with the baroclinic activity at midlatitudes that is particularly intense during winter when the meridional temperature gradients are stronger than during summer. The lag with respect to the perfect opposition of phase, observed in both the NH and the SH, which is of about 1 month in terms of time (Grinsted et al., 2004), is likely associated with the larger thermal inertia of the oceans when compared with land surfaces as it is for the NH Pacific Ocean and SH midlatitudes. This is supported by a comparative analysis of the annual component of the time series showing for the NH midlatitudes (dominated by land continents) an almost full opposition of phase between surface temperature and baroclinicity, and for the ocean regions a shift of about 1 month with respect to the phase opposition.

At the semiannual frequency, a phase shift of about 50° is observed in the SH and about 80° in the NH Pacific, with surface temperature delaying by about 1 month or more compared to the opposition of phase: results seem in agreement with the SAO phenomenon and may be indicative of the role of the semiannual harmonic in modulating NH synoptic time-scale baroclinic eddy activity (an example is the midwinter suppression characterizing the North Pacific storm tracks). Also, the high values of coherency found at the semiannual frequency suggest that, as it is in the SH, the contribution of such harmonic to the NH ocean variability is not negligible.

Bivariate spectral analysis is applied to CMIP3 model outputs by considering the common time section 1900–1999 and the same number of tapers as for the reanalysis ($K = 5$). The latter choice implied the same degrees of freedom and, hence, the same confidence level estimate. Results are shown in Figures 8–9.

With regard to the coherence spectra (Figures 8a, 9a), at the annual and semiannual frequencies, all CMIP3 models show high values well above the confidence level threshold both in the Pacific and the SH, in agreement with ERAI. Phase spectra (Figures 8b, 9b), instead, show some discrepancies with the reanalysis and among models (Table 3). Generally, in both hemispheres, there is more uncertainty in the reproduction of the phase of the semiannual cycle than of the annual one. In particular, while at the annual frequency almost all CMIP3 models reproduce quite well the opposition of phase with 1-month delay of surface temperature (exceptions are CGCM3.1 and ECHAM5/MPI-OM for the NH Pacific and GFDL-CM2.1 for the SH), at the semiannual frequency three (NH Pacific) and two (SH) out of six models display phase values within the reanalysis error intervals (see Table 3). However, it is worth noticing that no CMIP3 model reproduces the reanalysis values at both the annual and semiannual frequencies in the two hemispheres. Moreover, it emerges that larger errors characterize phase estimates at the semiannual frequency when compared to those obtained for the annual one, likely due to the less power of the six-month peak as resulted from the power spectra (Figure 5, 6).

To better illustrate the discrepancy among models, the scatter plots of the relative phase between surface temperature and baroclinicity at the annual and semiannual frequency, for both the NH and the SH, are shown in Figure 10a, b, respectively, for ERAI (green) and CMIP3 (blue). On the same plots results obtained from CMIP5 are also displayed (red) for comparisons, see also Table 3 for phase values. In performing the bivariate spectral analysis for CMIP5, the same time period used for CMIP3 has been considered (i.e., 1900–1999), with the exception of CanCM4 model having a different time section (1961–1999). It clearly emerges that for CMIP3 while the uncertainty in the phase at the annual frequency is bounded between about $\pm 15^{o}$ around an average value of about $220^{o}$, phase estimates at the semiannual frequency span a wider range of values (from $240^{o}$ to $360^{o}$ in the NH Pacific, from $120^{o}$ to $300^{o}$ in the SH). Findings seem to suggest that there is no evidence of the relationship between models' horizontal resolution and their performance. Furthermore, it is worth noting that for the semiannual frequency model errors are larger when compared with those obtained for the annual one (twice or more).

Results obtained for the most recent multi-model dataset CMIP5 show several improvements compared with CMIP3, especially for the semiannual frequency in the NH: phase values span between about $220^{o}$ and $300^{o}$ (i.e., about 1 month phase shift) for NH Pacific, and between about $180^{o}$ and $280^{o}$ (i.e., about 2 months) for the SH. Larger errors of model estimates allow their overlap with the reanalysis error bars; the non-overlapping among some models errors also occurs, denoting a degree of uncertainty in model data also in CMIP5. It is worth noticing that recently Di Biagio et al. (2014) evaluated whether CMIP3 and CIMIP5 models predict future shifts in the global baroclinic eddies and planetary scale wave activity, and found no significant improvements with CMIP5 ensemble and limited changes of baroclinic activity in RCP.4.5 scenarios. Differently, in the present analysis CMIP5 ensemble shows a better representation of the annual/semiannual cycles when compared with the previous version CMIP3.

Moreover, although MIROC5 show improved results, at the stage of the present analysis, overall, it is not possible to determine whether model changes, involving for example the horizontal resolution, have a significant impact on the representation of the semiannual period variability. A comprehensive evaluation of AOGCMs' performance in terms of capability to reproduce the observed annual and semiannual cycles should be the natural extension of the present study. At

the stage of the present analysis, it is not possible to identify specific indications for modellers to improve the model accuracy or major details concerning selected phenomena like SAO. Comparisons with additional reanalysis data or observations, as well as sensitivity studies (for example those concerning model resolution or parameterization schemes) should be carried out; for a given aspect to be analysed a set of model experiments should be carried out and inter- and intra-model comparisons taken into account.

Furthermore, to avoid any effect due the record length, we have repeated the analysis by considering a common record length for ERAI and CMIP3/5 (i.e., 32 years). As expected, results (not shown) display substantial agreement with no significant discrepancies.

As a final investigation, we have compared present results with those obtained using six models from the Atmospheric Model Intercomparison Project (AMIP), which uses observed distributions of sea-surface temperature and sea ice as

boundary conditions (see http://www-pcmdi.llnl.gov/projects/amip/index.php and model documentation), and ERA-20CM, a twentieth-century atmospheric model ensemble developed by the European Centre for Medium-Range Weather Forecasts (ECMWF, Hersbach et al., 2015). For AMIP runs (CanCM4, FGOALS-g2, GFDL-CM3, INM-CM4, MIROC5, MPI-ESM-MR) we have considered the common time section 1979–2009, while for ERA-20CM the period 1979–2011. Scatter plots of the relative phases are shown in the Figure 11 (ERAI in green, AMIP in magenta, ERA-20CM in blue, CIMIP5 in red).

Results appear comparable with those obtained with CMIP5, with a general slight improvement at the annual frequency ($10^o$–$15^o$). Using AMIP, small improvements (about $15^o$) are obtained for the Pacific sector at the semiannual frequency, while the model INM-CM4 shows a smaller relative phase in SH midlatitudes when compared with other models. At the stage of the present analysis, results suggest that the impact of observed SST on the modelled relative phase is primarily on the annual cycle (though limited to a few degrees) and, as expected, on the NH Pacific ocean sector.

Results for ERA-20CM appear consistent with those obtained for the reanalysis ERAI.

## 4 Summary and conclusions

The annual and seasonal cycles in the time series of 2-meter temperature and baroclinicity are analyzed in both hemispheres using ERAI reanalysis data covering the period 1979–2015 and AOGCM outputs from CMIP3 and CMIP5 experiments of different record lengths.

The baroclinicity index time series is estimated through the tropospheric meridional temperature gradient (Eady maximum growth rate), which is computed as the difference between the poleward and equatorward edges midlatitude band ($30^o$–$60^o$ for the NH and $30^o$–$70^o$ for the SH, respectively), and zonally and vertically averaged in each hemisphere, while near-surface

temperature time series are obtained by zonally and meridionally averaging over the same latitude band. After a preliminary analysis of the NH zonally averaged fields, we focus on the NH Pacific sector, where a more pronounced and statistically significant peak at the semiannual frequency is found for both 2-meter temperature and baroclinicity.

The spectral analysis carried out applying the MTM method shows that annual and semiannual periodic components are the dominant features in the reanalysis time series and model outputs. In particular, results show the occurrence of the semiannual oscillation in the zonally averaged baroclinic activity in both hemispheres. The presence of the semiannual peak in the NH baroclinicity may be considered a signature of the midwinter Pacific suppression as suggested by Nakamura et al. (1992), but further investigations are necessary.

The bivariate analysis of coherency and phase spectra between baroclinicity and surface temperature show discrepancies between ERAI and model data, as well as among models, especially for the semiannual frequency. In particular, for ERAI it is found that:

(i) At the annual and semiannual frequencies, a very high coherency between the two selected variables is observed;

(ii) At the annual frequency, in both hemispheres baroclinicity leads surface temperature of about $30^o$ with respect to the phase opposition;

(iii) At the semiannual frequency, the relative phase is shifted by about $70^o$ with respect to the opposite phase condition in the NH Pacific and by about $55^o$ in the SH (i.e., about 1 month delay of surface temperature with respect baroclinicity).

For models outputs:

(iv) For what concerns the coherency spectra, at the annual and semiannual frequency, coherency is well represented by all models in both hemispheres and well exceeds the confidence level. Results for CMIP5 models are in agreement with those presented for CMIP3;

(v) At the annual frequency, phase estimates in both hemispheres are bounded between about $\pm15^o$ around an average value of $220^o$ (i.e., about 1 month phase shift). At the semiannual frequency, model relative phases between surface temperature and baroclinicity show wider dispersion in both hemispheres, denoting discrepancies among models and wider uncertainty in the estimates. Results for CMIP5 display improvements when compared with CMIP3 with a reduction of the discrepancy among models, especially in the NH Pacific. Larger errors found at the semiannual frequency make phase estimates of most models consistent with the reanalysis (i.e., they fall within the ERAI confidence interval) but discrepancies still occur especially in the SH.

In performing the present analysis two assumptions have been made that might be considered restrictive with possible impacts on the obtained results. They are the vertical averaging, which has not been weighted for the mass of the atmosphere, and the choice of the monthly mean of daily means data. About the former, it has been observed that the discretization of the vertical pressure levels already accounts for that, while the impact of the latter choice has been verified against the results obtained using daily means with not statistically significant differences.

Furthermore, the choice of the maximum Eady growth rate as an index of baroclinicity has been considered suitable for the purpose of the present paper because it has been widely used in the international literature and, in the framework of zonally averaged atmosphere, the variability of the tropospheric static stability (unlike the static stability ratio between troposphere and stratosphere) has been found to have a secondary effect on baroclinic activity (Bordi et al., 2002; Fantini, 2004). It is worth noticing that the presence of a statistically significant semiannual peak in near-surface temperature spectral estimates, may suggest that the internal forcing exerted by baroclinic eddies play a role in modulating the annual cycle. The existence of the semiannual period in both near-surface temperature and baroclinicity index, in fact, might be related to the result of a feedback mechanism between baroclinic activity and near-surface temperature through the effects of the eddies heat transports (i.e., their impact on the meridional temperature gradients), in analogy with what happens in SAO phenomenon. According to recent arguments on the constraints to the applicability of the linear Ruelle response theory to the climate system (Lucarini, et al., 2016), this leaves open the possibility that the six-month modulation might result from the non-linear response of near-surface temperature and baroclinic activity to the external solar forcing depending on time scales and regions considered.

To be noted is that the six-month modulation of the baroclinicity index is somewhat not surprising, since the relationship between baroclinic activity and SAO has been widely investigated (e.g., Walland and Simmonds, 1999). It has been found that the SAO phenomenon is related to the half-yearly wave in the meridional temperature gradient at high southern latitudes that implies seasonal fluctuations of baroclinicity and surface pressure; moreover, the variation of the static stability during the year seems to modulate the efficiency of baroclinic conversion. Some evidences of a semiannual modulation have been found also in the NH at regional scale (e.g., Wikle and Chen, 1996), suggesting a mechanism for the SAO in the NH based on the east-west land-sea contrast, similarly to the north-south differential heating in the SH proposed by van Loon (1967). Present results support the existence of such oscillation in the NH Pacific region; it is left to a future study whether it is just the projection on the zonal average of regional scale processes or it is the signature of a global scale phenomenon.

Present results suggest a reduction in model discrepancies from CMIP3 to CMIP5 in the NH. It is worth noticing that this is not supported by evidences about the intensity of the atmospheric meridional heat transports as suggested by Lucarini et al. (2014), although some improvements are found on the position of the SH peak in CMIP5 when compared to CMIP3.

Present findings contribute to better characterize the cyclic response of current global atmosphere-ocean models to the external solar forcing that is of particular interest for seasonal forecasts. Knowing whether the semiannual cycle characterizes or not the climate variability of a given midlatitude region, in fact, may be useful to better verify (and eventually improve) the skill of seasonal forecasts. The annual and seasonal cycles are also important modulations of El Niño Southern Oscillation (ENSO), which is certainly the dominant driver for seasonal prediction (e.g., Troccoli 2010). The discrepancies emerged with reanalysis and among models, at least for the AOGCMs here considered, in properly reproducing the seasonal cycle of surface temperature and baroclinicity require further investigations. A larger set of models should be considered and the performance of a model ensemble against reanalysis should be assessed rather than that of a single model. In doing this, a simple metric for the seasonal cycle should be developed and tested (see for example Gleckler

et al., 2008). Finally, in light of the results here discussed, the study of the possible teleconnection between the intertropical seasonal variability and midlatitude circulation, as suggested for example by Vimont et al. (2001), would be interesting to deepen.

**Acknowledgements**

We acknowledge the modeling groups, the Program for Climate Model Diagnosis and Intercomparison (PCMDI) and the WCRP's Working Group on Coupled Modelling (WGCM) for their roles in making available the WCRP CMIP3 and CMIP5 multi-model datasets. Support of this dataset is provided by the Office of Science, U.S. Department of Energy.

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

**Table 1: Main characteristics of CMIP3 and CMIP5 models used for the analysis. Concerning the vertical resolution, "C" refers to cubed-sphered resolution, "Z" to σ-pressure hybrid coordinates.**

| CMIP3 | Model | Horizontal resolution (atmosphere) | Vertical resolution (atmosphere) | Horizontal resolution (ocean, in lon × lat) | Simulated years |
|---|---|---|---|---|---|
| High resolution | MIROC3.2 | T106 | σ56 | $0.28125^o$x$0.1875^o$ | 1900–2000 |
| Medium resolution | CGCM3.1 | T63 | Z31 | $1.40^o$x$0.94^o$ | 1850–2000 |
| | ECHAM5/MPI-OM | T63 | H31 | $1.5^o$x$1.5^o$ | 1860–2009 |
| | FGOALS-g1.0 | $2.8^o$x$2.8^o$ | σ26 | $1.0^o$x$1.0^o$ | 1850–1999 |
| | GFDL-CM2.1 | $2.5^o$x$2.0^o$ | H24 | $1.0^o$x$1.0^o$ | 1861–2000 |
| Low resolution | INM-CM3.0 | $5^o$x$4^o$ | σ21 | $2.5^o$x$2.0^o$ | n/a |
| | | | | | |
| **CMIP5** | | | | | |
| High resolution | MIROC5 | | Z40 | $1.4^o$x0.5 ($1.4^o$) | 1850–2012 |
| Medium resolution | CanCM4 | T63 | Z35 | $1.41^o$x$0.94^o$ | 1961–2005 |
| | MPI-ESM-MR | T63 | Z47 | $1.5^o$x$1.5^o$ | 1850–2005 |
| | FGOALS-g2 | $2.8^o$x$2.8^o$ (equal area) | σ26 | $1.0^o$x$1.0^o$ | 1850–2005 |
| | GFDL-CM3 | C48 | Z48 | $1.0^o$x$1.0^o$ | 1860–2005 |
| | INM-CM4 | $2.0^o$x$1.5^o$ | σ21 | $1.0^o$x$0.5^o$ | 1850–2005 |

**Table 2: Absolute phases of T2m and σ<sub>BI</sub> for the reanalysis ERAI and models (CMIP3 and CMIP5).**

| | *T2m* | | | | $\sigma_{AB}$ | | | |
|---|---|---|---|---|---|---|---|---|
| | Annual (NH) | Semiannual (NH) | Annual (SH) | Semiannual (SH) | Annual (NH) | Semiannual (NH) | Annual (SH) | Semiannual (SH) |
| **ERAI** | 148.68 | 273.59 | 335.56 | 343.02 | 2.88 | 172.22 | 190.00 | 217.6 |
| **CMIP3** | | | | | | | | |
| CGCM3.1 | 139.25 | 216.24 | 330.89 | 78.37 | 3.11 | 183.29 | 187.13 | 217.6 |
| ECHAM5 | 143.70 | 264.36 | 326.01 | 327.07 | 6.32 | 178.85 | 192.6 | 213.5 |
| FGOALS-g1.0 | 144.00 | 193.64 | 335.70 | 276 | 359.5 | 175.46 | 197.41 | 197.66 |
| GFDL-CM2.1 | 153.14 | 282.19 | 331.17 | 320.40 | 10.82 | 188.20 | 163.28 | 202.22 |
| INM-CM3.0 | 151.25 | 259.11 | 338.5 | 311.3 | 3.27 | 198.7 | 196.5 | 219.13 |
| MIROC3.2 | 146.03 | 283.37 | 328.62 | 315.9 | 0.92 | 179.81 | 223.40 | 239.80 |
| **CMIP5** | | | | | | | | |
| CanCM4 | 145.74 | 290.01 | 330.59 | 14.35 | 1.98 | 175.38 | 180.66 | 215.62 |
| FGOALS-g2 | 144.02 | 262.26 | 335.17 | 318.31 | 8.84 | 158.86 | 189.91 | 202.38 |
| GFDL-CM3 | 123.36 | 228.3 | 331.79 | 316.85 | 12.65 | 176.35 | 204.01 | 205.80 |
| INM-CM4 | 150.59 | 294.39 | 332.38 | 1.31 | 6.03 | 204.06 | 197.53 | 222.25 |
| MIROC5 | 140.95 | 261.60 | 325.38 | 310.26 | 358.87 | 178.02 | 182.64 | 208.35 |
| MPI-ESM-MR | 143.44 | 263.32 | 326.26 | 319.93 | 9.98 | 188.51 | 191.75 | 223.12 |

**Table 3: Phase values between surface temperature and baroclinicity computed using MTM method. Errors at 95 % confidence level are computed using Montecarlo simulations ($2^{13}$); phase values that fall within the confidence interval of the reanalysis ERAI are in bold.**

| | NH Pacific (30°–60°) | | SH (30°–70°) | |
|---|---|---|---|---|
| | *1 yr⁻¹* | *2 yr⁻¹* | *1 yr⁻¹* | *2 yr⁻¹* |
| ERAI | 213.1±3.3 | 258.0±11.2 | 214.5±6.9 | 235.4±10.8 |
| **CMIP3** | | | | |
| MIROC3.2 | **214.1±2.3** | **253.6±14.8** | **209.8±3.5** | 265.3±17.0 |
| CGCM3.1 | 224.3±3.1 | 328.4±24.6 | **215.8±2.7** | 140.8±19.3 |
| ECHAM5/MPI-OM | 221.2±2.3 | **274.7±17.0** | **225.3±5.1** | **245.5±21.9** |
| FGOALS-g1.0 | **215.8±2.5** | 338.0±19.4 | **222.4±2.6** | 281.4±16.3 |
| GFDL-CM2.1 | **216.7±2.7** | **264.1±19.8** | 191.4±4.5 | **244.0±17.6** |
| INM-CM3.0 | **212.1±2.4** | 303.8±26.6 | **216.8±2.9** | 268.7±18.3 |
| **CMIP5** | | | | |
| MIROC5 | **217.8±3.7** | **274.5±16.0** | **209.6±10.0** | **258.4±27.7** |
| CanCM4 | **216.7±2.9** | **244.4±10.8** | **209.0±4.5** | 198.8±17.9 |
| MPI-ESM-MR | 226.3±3.5 | **283.0±17.1** | **224.6±8.0** | **260.0±23.2** |
| FGOALs-g2 | 224.2±3.5 | **253.7±15.7** | **213.3±5.0** | **244.5±25.7** |
| GFDL-CM3 | **220.7±14.0** | **242.6±18.8** | 227.8±12.1 | 257.7±22.3 |
| INM-CM4 | **215.8±3.7** | **266.2±18.0** | 223.7±3.7 | 218.3±20.0 |

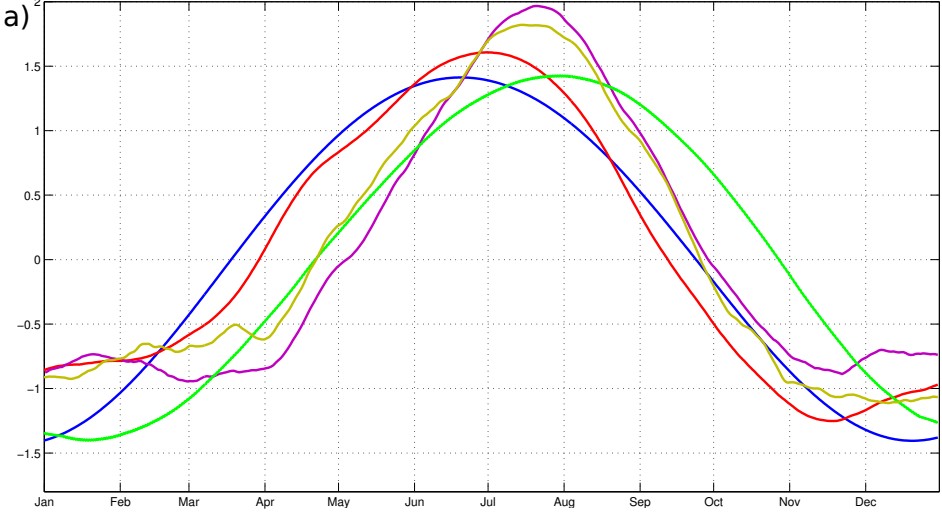

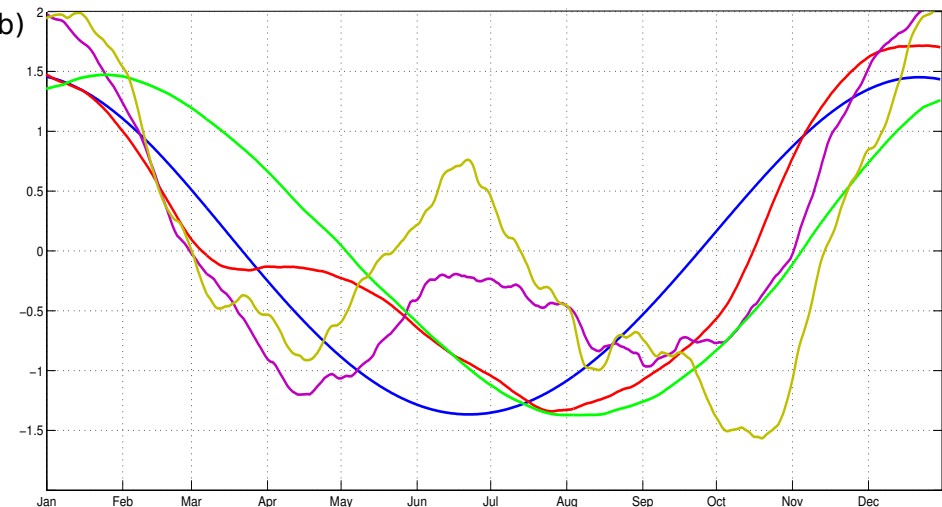

**Figure 1: Mean annual cycles of midlatitude ERAI fields for the Northern (a) and Southern Hemispheres (b): surface temperature (T2m, green), baroclinicity index ($\sigma_{BI}$, purple), solar radiation at the top of the atmosphere (TOA, blue), mean sea level pressure variance (MSLP, red), meridional gradient of the geopotential field at 500 hPa (khaki). Baroclinicity index, GPH, and MSLP variance cycles have been reversed so that their maxima occur in summer as for the other fields. Values are standardized.**

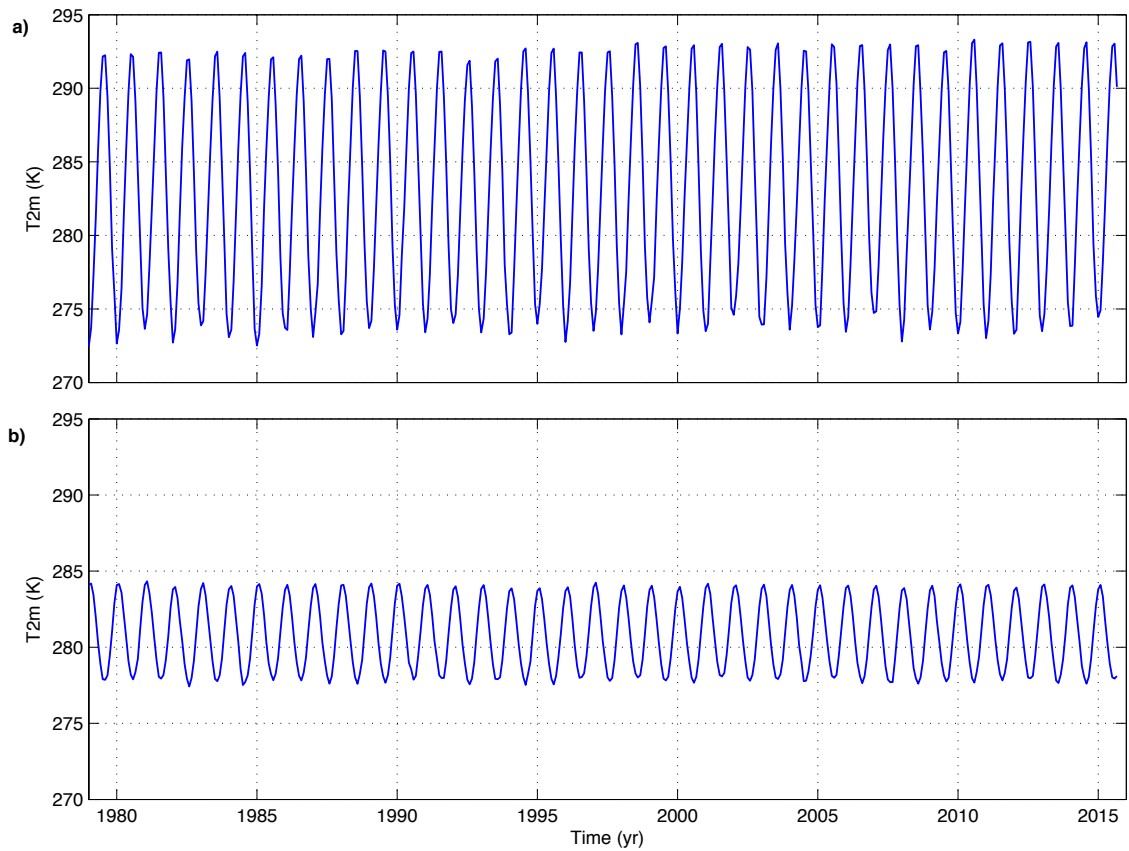

**Figure 2: Monthly mean time series (1979–2015) of ERAI surface temperature (T2m) averaged over: a) the Northern Hemisphere (NH) and b) Southern Hemisphere (SH) midlatitudes (30–60$^{o}$N and 30–70$^{o}$S, respectively). Units are degree K.**

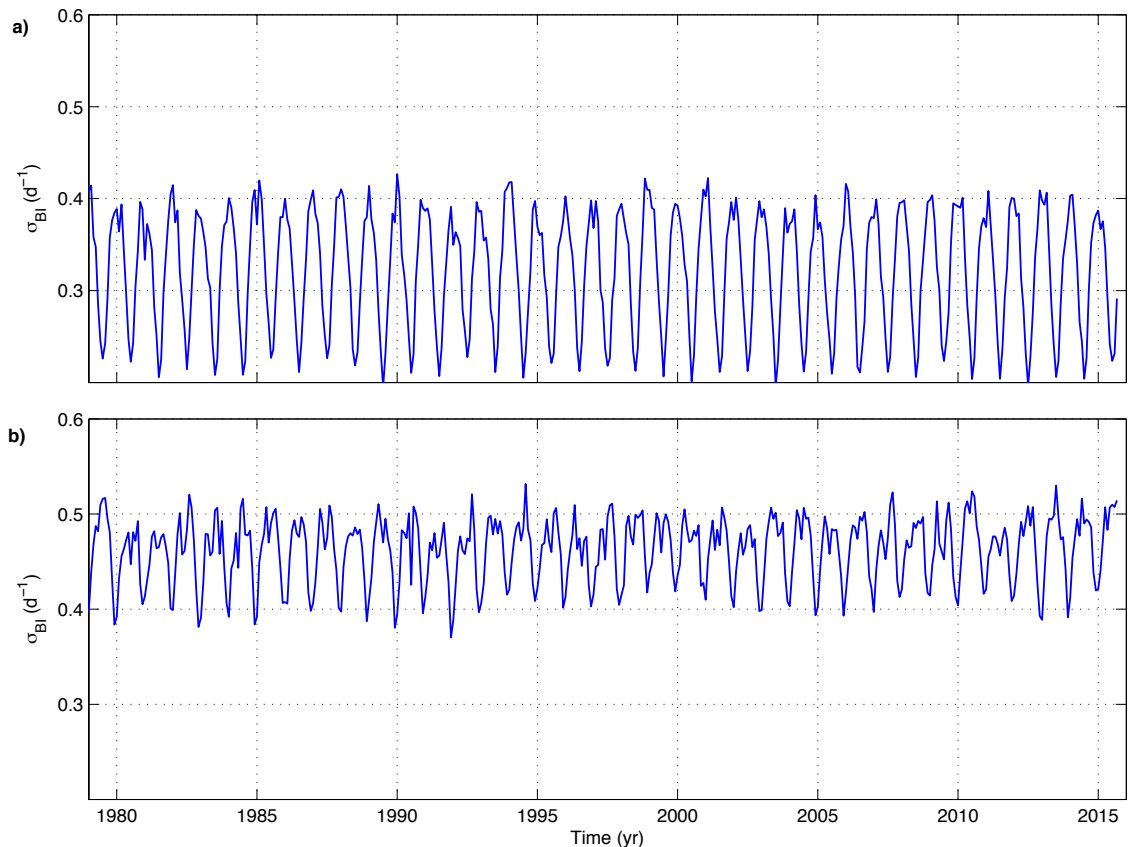

**Figure 3:** As in Figure 2 but for ERAI baroclinicity index ($\sigma_{BI}$). Units are $d^{-1}$.

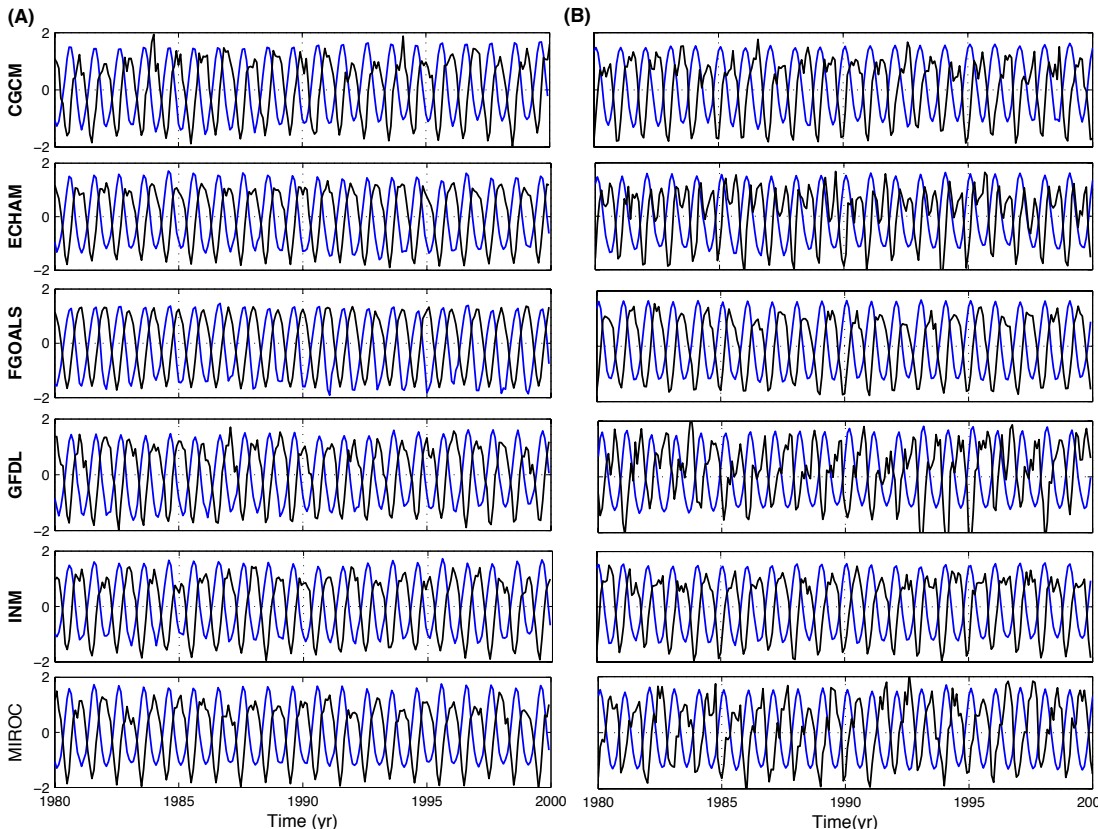

**Figure 4: CMIP3: Time series of surface temperature (T2m, blue) and baroclinicity ($\sigma_{BI}$, black) for the 6 selected AOGCMs for the common time period 1979–1999. Time series are standardized.**

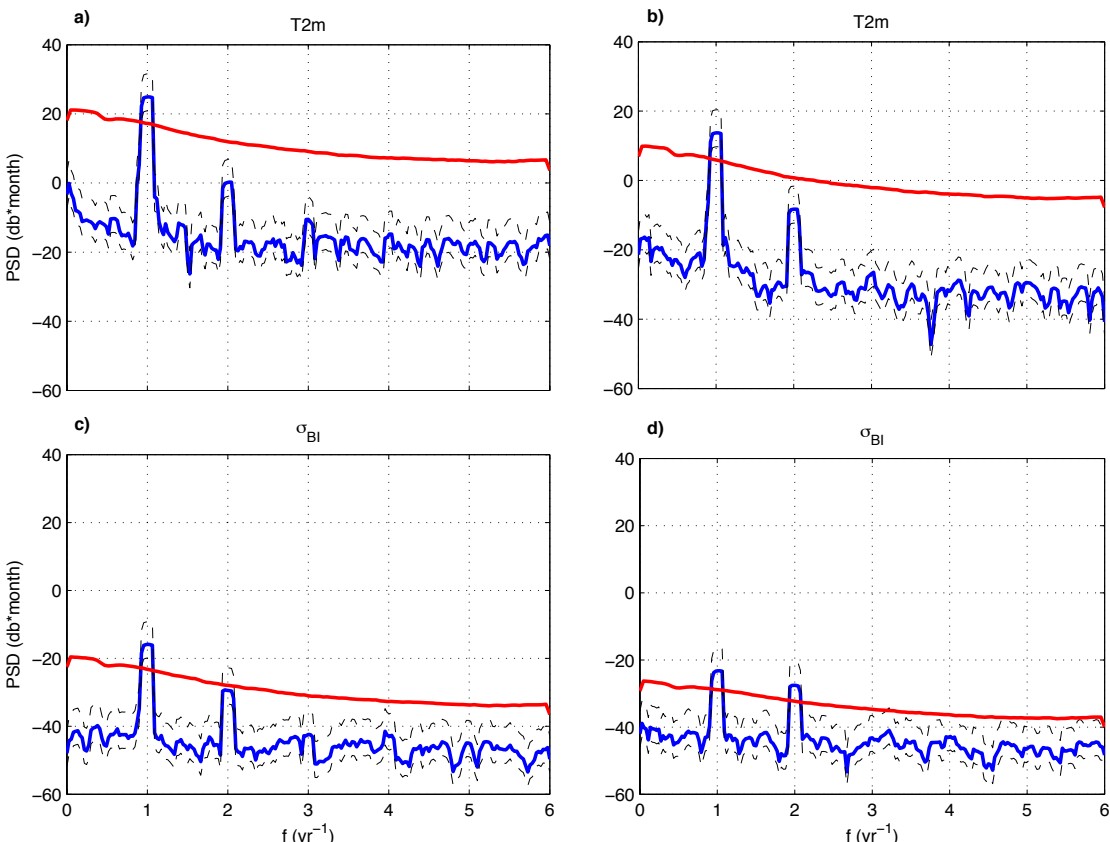

**Figure 5: Power spectral density (PSD) estimates of ERAI mean surface temperature (T2m) and baroclinicity ($\sigma_{BI}$), for the Northern (a, c) and Southern Hemispheres (b, d) midlatitudes, computed using the MTM method (parameters $p = 3$, $K = 5$). The 0.99 confidence intervals (dashed lines) are obtained by the Chi-squared distribution with 7 degrees of freedom; the red noise spectrum at 95 % level is in red. Frequency on the x-axis is in $yr^{-1}$.**

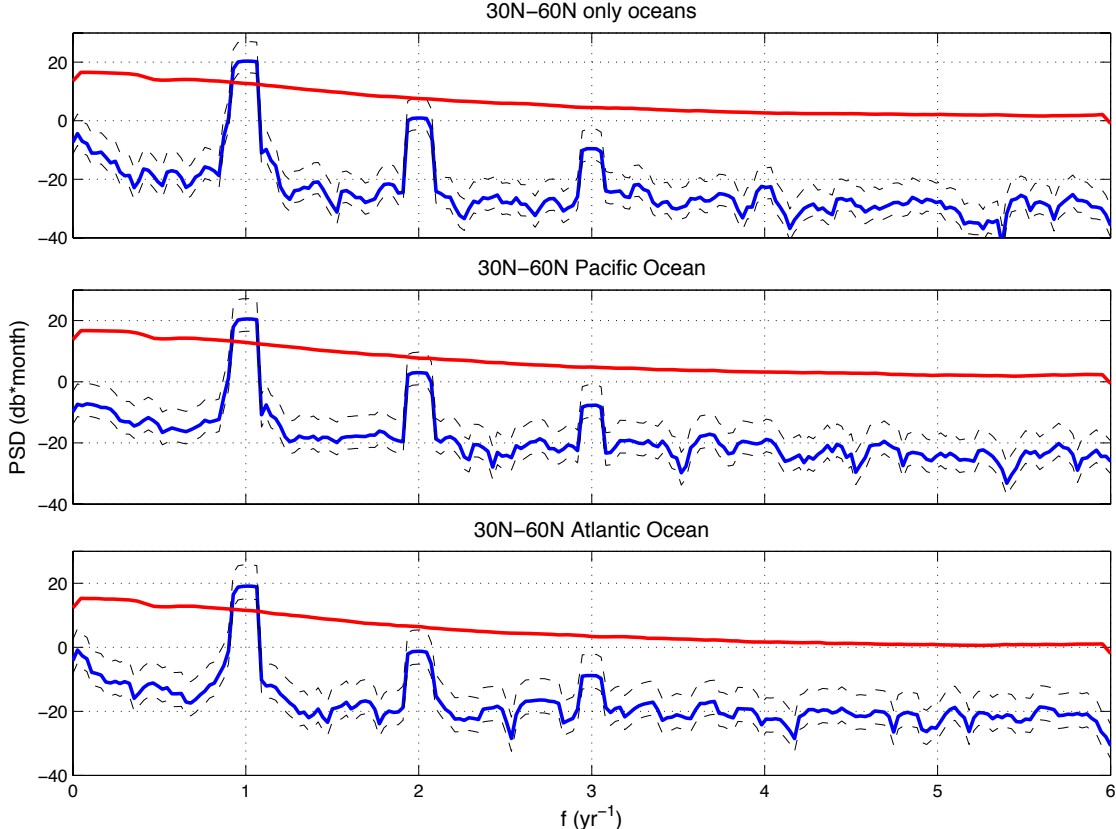

**Figure 6: As in Figure 5 but for T2m and the following ocean regions: NH oceans (30°–60°N), Pacific Ocean (30°–60°N), Atlantic Ocean (30°–60°N).**

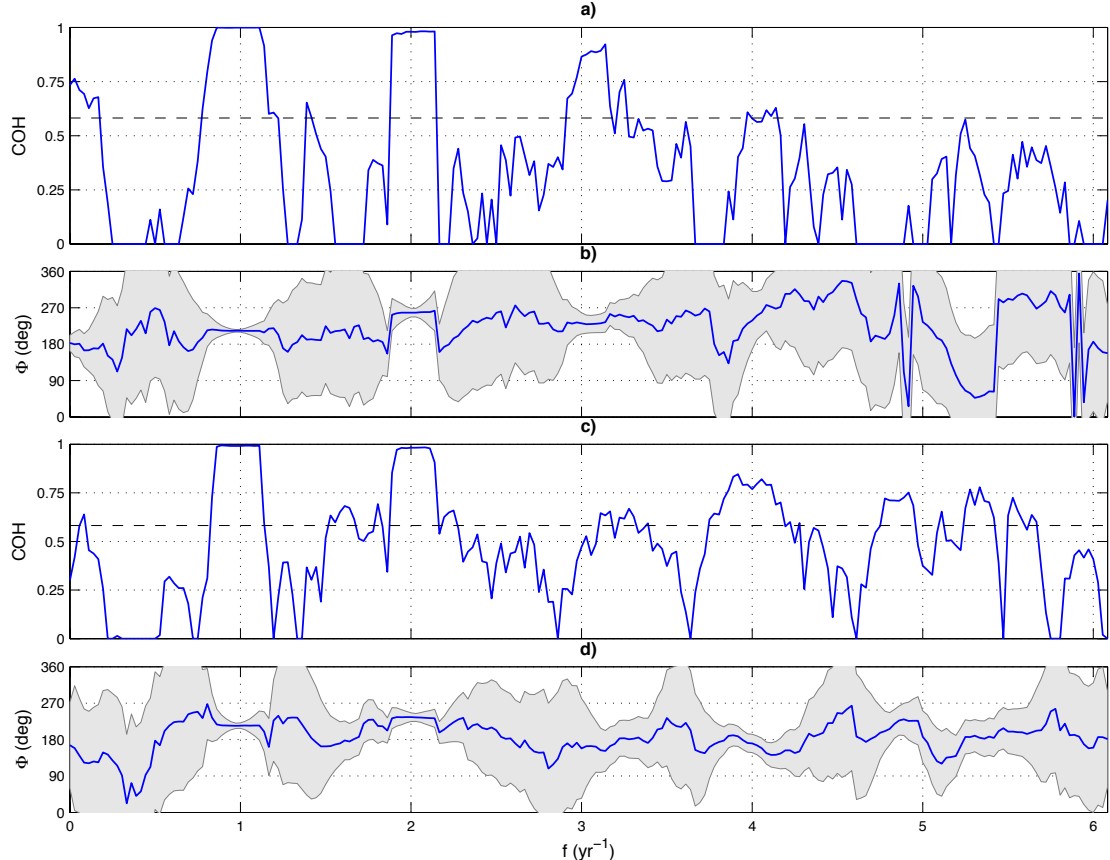

**Figure 7: Bivariate spectral analysis (coherency and phase spectra) of ERAI surface temperature and baroclinicity for the period January 1979–September 2015: a)–b) NH midlatitude band 30°–60°N over the Pacific Ocean; c)–d) SH midlatitude band 30°–70°S. Dashed horizontal line in the coherency plot represents the 97.5% significance level; shaded areas in the phase plots represent the 95% level of significance obtained by means of $2^{13}$ Montecarlo simulations. Frequency on the x-axis is in $yr^{-1}$.**

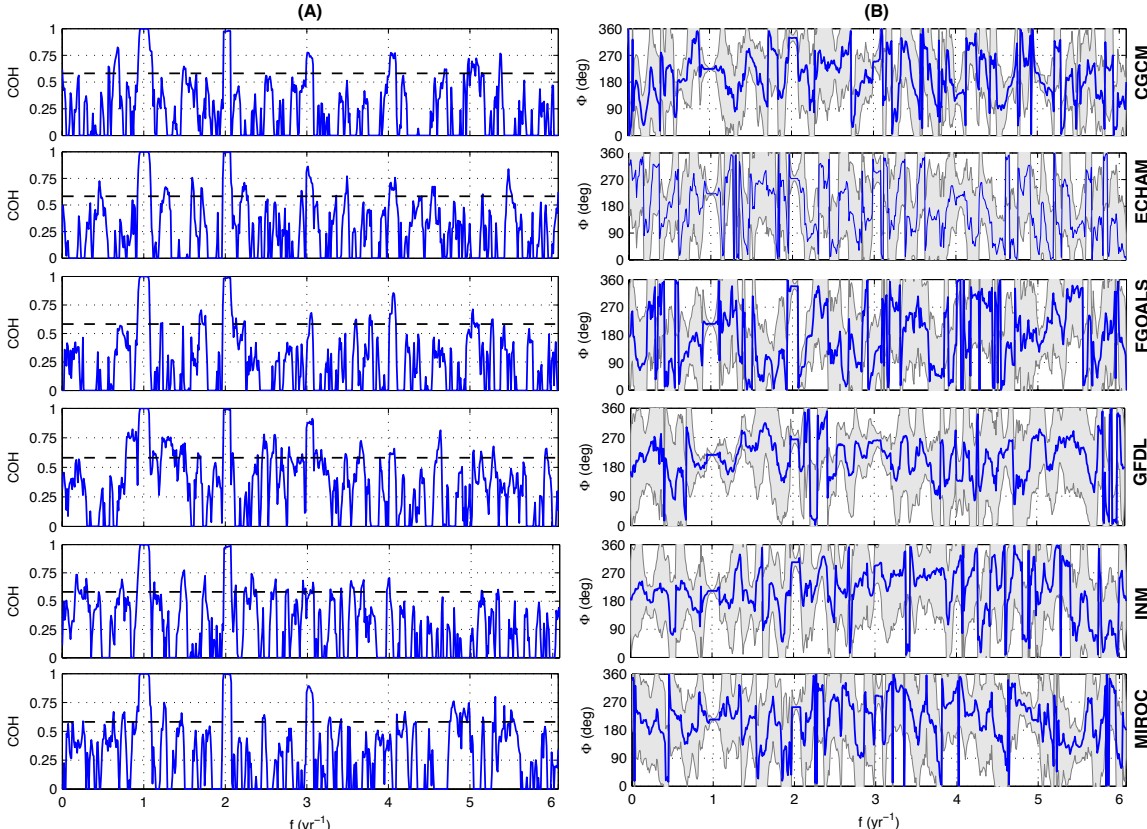

**Figure 8:** **Bivariate spectral analysis (coherence and phase spectra) of surface temperature and baroclinicity for the 6 selected CMIP3 models computed over the time period 1900–1999. Plots refer to the NH midlatitude band $30^o$–$60^o$N over the Pacific Ocean. Dashed horizontal line represents the 97.5% significance level; shaded areas represent the 95% level of significance obtained by means of $2^{13}$ Montecarlo simulations.**

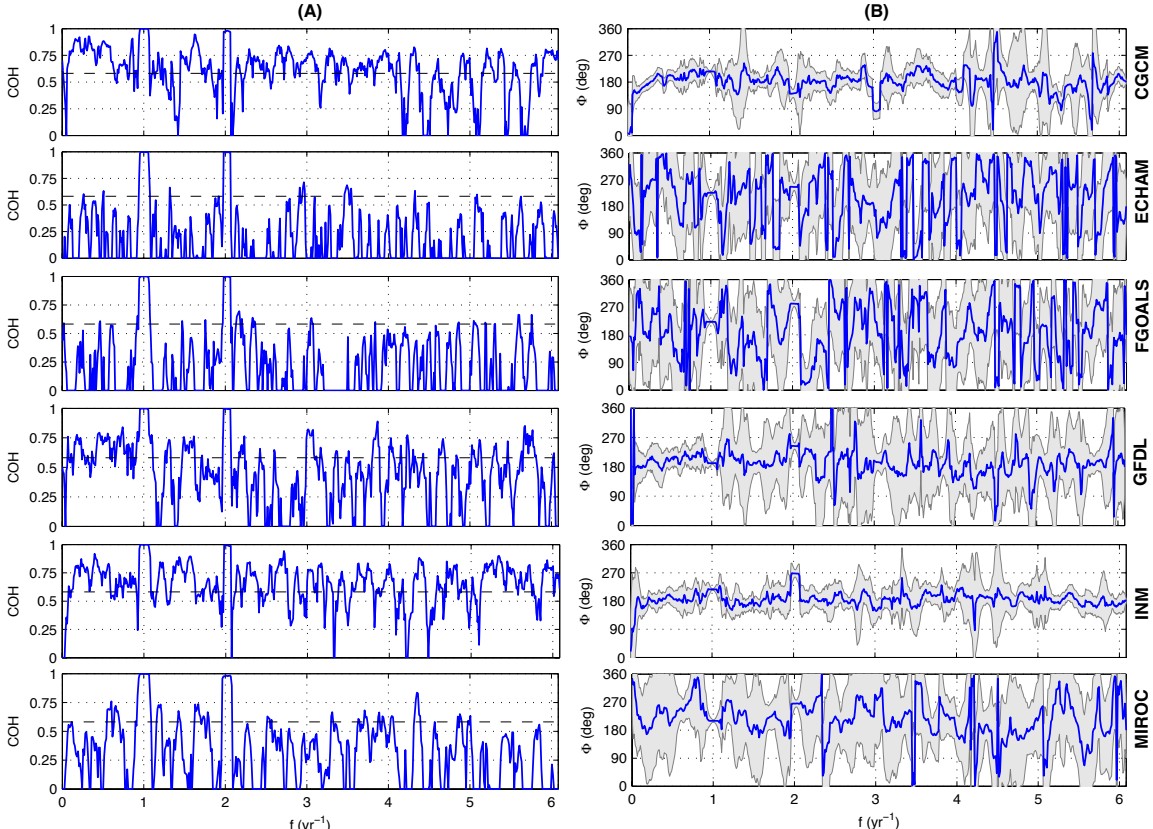

**Figure 9: As in Figure 8 but for the SH midlatitude band 30°–70°S.**

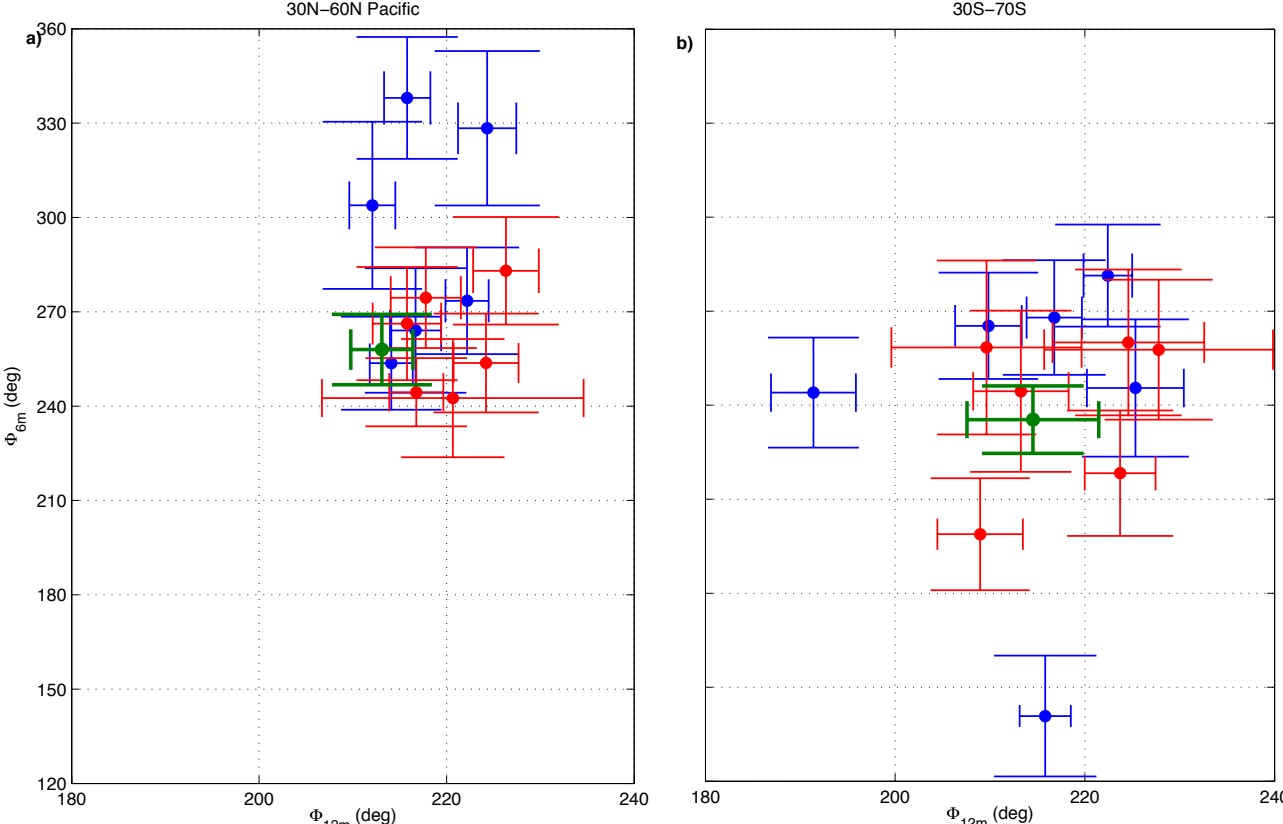

**Figure 10: Scatter plots of the relative phase between surface temperature and baroclinicity at the annual ($\Phi_{12m}$) and semiannual ($\Phi_{6m}$) frequency for: a) NH Pacific Ocean $30^o$–$60^o$N, b) SH latitude band $30^o$–$70^o$S. Estimates for ERAI reanalysis are in green, for CMPI3 and CMIP5 models are in blue and red, respectively. Error bars have been computed as in Figure 7.**

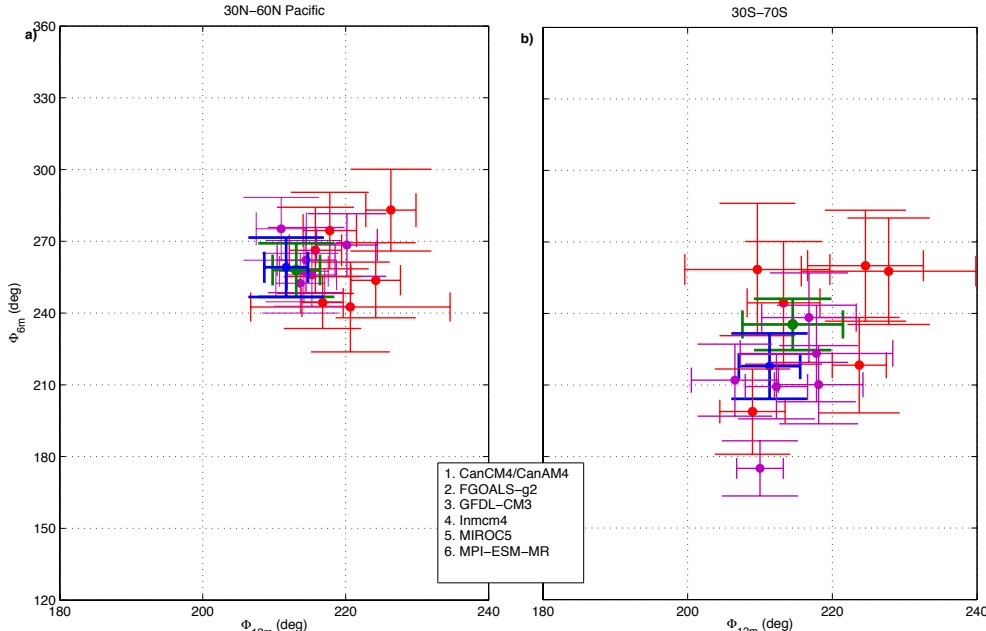

**Figure 11: Scatter plots of the relative phase between surface temperature and baroclinicity at the annual ($\Phi_{12m}$) and semiannual ($\Phi_{6m}$) frequency for: a) NH Pacific Ocean $30^o$–$60^o$N, b) SH latitude band $30^o$–$70^o$S. ERAI is in green, AMIP in magenta, ERA-20CM in blue, and CMIP5 in red.**

