# Peer review of "Annual and semiannual cycles of midlatitude near-surface temperature and tropospheric baroclinicity: reanalysis data and AOGCMs simulations"

_Earth System Dynamics, 2016_

## Referee Comment (RC1) · Anonymous Referee #1 · 3 Aug 2016

The authors investigate the annual and semiannual cycles of atmospheric near surface temperature and baroclinicity (maximum Eady growth rate) in midlatitudes. They analyze the statistical relationship between the two quantities, and assess the ability of CMIP3 and CEMIP5 models to reproduce properties derived from ERA Interim reanalysis. The results show high coherence between the two variables for both the annual and the semiannual cycle, but with different relative phases. The CMIP models show good agreement with reanalysis for coherence at annual and semiannual frequency. For relative phase at semiannual frequency larger differences between models and reanalysis and among the models are observed. Improvements for CMIP5 models compared to CMIP3 are found.

[Figure]

General To test the ability of climate models to simulate the present day climate, and, thus, to give some confidence in their projection of potential future climates the simulated annual and semiannual cycles are appropriate testbeds. In addition, near surface temperature and baroclinicity (or maximum Eady growth rate, as an indicator for eddy activity) are important quantities defining the climate state. Thus, the study conducted here addresses relevant scientific questions and fits the scope of Earth System Dynamics. The methodology applied is sound and overall presentation is well structured and clear. Though the paper does not present novel concepts, ideas, tools or data, I think that it presents potentially valuable new results. However, I have three specific points the authors need to address before I can recommend the paper to be accepted.

Specific 1) I appreciate the spectral analysis of near surface temperature and baroclinicity individually, but the significance of doing the spectral coherence/phase analysis between both as presented in this paper is not clear to me. The authors only give vague motivation for doing the coherence analysis by saying that '... temperature is taken as a proxy in climate change studies' (P2L18) and '... investigating the possible relationship...may help to better understand the role played by the latter...' (P3L16+17). For me, the combined analysis of near surface temperature and baroclinicity appears a bit arbitrary and the meaning/interpretation of the results seems little conclusive:

(i) The results seem to indicate that the coherence is mainly restricted to the annual cycle (and higher harmonics) which may simply because both variables have an annual cycle without any physical relation between them. At the moment, I cannot see a significant indication for a relation other than a pure statistical one (see also 2). If understanding of the relation between near surface temperature and baroclinicity is an aim of this study, there need to be some more statements/discussions on this in the conclusion. At the moment the paper gives some discussions about baroclinicity and temperature gradients (e.g. SAO), but the link to temperature itself is missing, or appears to be an indirect one which is related to changes in near surface temperature gradients. In other words: for analyzing a possible relation between near surface tem-

perature and baroclinicity the (equator-to-pole) temperature gradient may be a more obvious quantity. Why do the authors not consider this directly?

(ii) A second aim of the study is the evaluation of CMIP models. Given the somehow unclear (physical) interpretation of the observed coherences and relative phases between baroclinicity and temperature (see above), looking at the absolute phases of both variables may be of significant benefit. This may also help to identify the source of the discrepancies with respect to the semiannual cycle: do the phases of the baroclinicity or the near surface temperature (or both) differ? In this respect: L12 of the abstract suggests that also the absolute phases are considered.

2) P12L11-13: The authors state that '... the presence of a statistically significant semiannual peak in surface temperature spectral estimates, may suggest that the internal forcing exerted by baroclinic eddies play a role in modulating the annual cycle'. I do not understand this statement: Why does a significant semiannual peak observed in the surface temperature indicate a forcing different from the solar one, and, moreover, a forcing related to baroclinic eddies? Please clarify.

3) In the paper the authors use surface temperature and near surface temperature as synonymously, but actually atmospheric near surface temperature is used. Thus, to be more precise/clear, the authors may use the term near surface temperature throughout the paper. I do not think that the results are very sensitive to the particular choice of surface or near surface temperature. However, there are major differences between both variables, e.g. while the surface temperature is prognostic in the models, the near surface temperature is diagnosed from the lower atmospheric levels by using a somehow artificial role related to the computation of the surface fluxes.

4) The summary and conclusions section suggests that zonally (and vertically) averaged values are used in this study (P11L7). But, from the results section we learn that for the northern hemisphere the analysis focus on the Pacific. This should be made clearer (more consistent) in the summary and conclusions (Otherwise, e.g., the origins

of results (iii) and (v) are unclear for readers focusing on the summary and conclusions only).

---

## Referee Comment (RC2) · Anonymous Referee #2 · 10 Aug 2016

**Review of the paper "Annual and semiannual cycles of midlatitude surface temperature and baroclinicity: reanalysis data and AOGCMs simulations", by V. Lembo, I. Bordi and A. Speranza.**

This paper analyses some statistical properties of the global atmosphere in comparison with results obtained from general circulation/climate models. In particular, the annual and semiannual cycles of near-surface temperature and of an index of baroclinic activity are examined in detail.

The technical execution is basically correct and applied analysis methods are reasonably funded and appropriate. At least part of the results is original and interesting. What is basically missing, in my opinion, is a deeper physical discussion of the working hypotheses and of the results. This deficiency makes this paper to appear more as a preliminary contribution than a conclusive one (the authors are aware of that, as they write in several circumstances "... require further investigation" or so). I think that this paper would gain in quality if a discussion is provided at some appropriate points and in the conclusions, as expressed in the following general comments. Therefore, I consider that this paper can be made worth of publication if a revision will be made in response to my indications given below.

General comments.

1. Two parameters are chosen for the analysis, the near-surface temperature (T2m) and a tropospheric-averaged baroclinicity index ($\sigma_{BI}$), expressing a theoretical grow rate of baroclinic instability.  I think that this choice of the parameters should be better explained. In fact, the physical link between such quantities is very complex and can be very different depending on the time scale of the processes involved. The fact that they are strictly correlated (actually, anti-correlated) in the annual cycle is quite obvious, with reference, for example, to the results exposed in the paper by Donohoe and Battisti (2003, cited in this paper): considering the large impact on the atmospheric seasonal heating due to direct absorption of the solar radiation (not neglecting, of course, the role of land and ocean fluxes), and considering the very large annual cycle of both solar radiation and of the meridional gradient of it (which even changes it sign between solistices), it is not surprising that both quantities are strongly modulated by the astronomical cycle. However, this does not mean that the two quantities have a simple relation between each other, stemming from the atmospheric circulation dynamics! It seems that the choice of T2m has been made having in mind the application of climatic models to the long-term global changes (see for example line 18 of page 2), but this is something different form considering the seasonal-annual variability. On the other hand, the baroclinic index may be a subtler indicator of long term variability, while it is closely related to many aspects of the general circulation dynamics, including of course mid-latitude baroclinic "turbulence" and its limiting factor.

2. Concerning the evaluation of the AOGCMs, the intercomparison based on seasonal-annual harmonics may not be very appropriate to assess model performance with respect to long-time variability, associated for example with the dynamics of slow processes like those related to ocean deep circulation, evolution of the cryosphere and the biosphere etc. However, I agree that a "good" climatic model should behave well also in simulating/predicting intra-annual time scales.

3. Regarding the appreciation of annual vs semiannual cycles, one should consider, in principles, that while the astronomical forcing is basically annual ta mid latitudes, it is

basically semiannual at the equator. The analysis of the paper is restricted to the mid latitudes, but how does the intertropical semiannual variability affect the extratropical variability? The authors correctly mention possible relations with the SAO, but a more specific discussion (at least more detailed references to the literature) is needed in this respect, regarding SAO and perhaps more general semiannual aspects of the atmospheric global circulation.

4. As anticipated above, the discussion of the results and the conclusion miss a sufficient consideration of the physical implications in terms of known properties of the general circulation of the atmosphere. I am asking for at least a somehow better indication of the perspectives derived from the results of this paper in terms of (a) possible relationships between the ERAI-based results and some known aspects of the global atmospheric dynamics (for example the SAO), and (b) somehow less vague indications of the possible implications concerning AOGCMs' performance, so as to provide modellers with more specific hints (only the horizontal resolution is mentioned at line 33 of page 10, which is perhaps not the most important aspect: I assume that many more subtle physical and dynamical problems have still to be solved to improve the model accuracy).

Minor comments.

1. Title (and also in the text: for example, at line 9 page 1, lines 21 and 24 page 2): it is not clear (before reading the definition in Sect. 2.2) if the word "surface" is referred also to "baroclinicity". So I suggest to use "tropospheric baroclinicity" or something similar in the tile and in the text, before the exact definition is given in Sect. 2.
2. Page 2, line 2: contribute.
3. Pls. specify that "seasonal heating" is intended as the heating variability after subtracting the annual mean.
4. Page 2, line 8: "... and the atmosphere is heated from below": this seems to contradict the previous sentence that most of the (seasonal) atmospheric heating is due to direct atmospheric absorption. Pls. clarify.
5. Page 2, line 17: perhaps "... the first efforts *in understanding* the climate impact...".
6. Page 2, line 18: proxy of/for what? of globally averaged temperature? Pls. specify.
7. Page 2, line 19: "*its* key role".
8. Page 2, line 24: AOGCM in place of GCM (as elsewhere in the text).
9. Page 3, lines 2-3: consider this change "there are regions of enhanced eddy activity ("storm tracks"), where... decay.".
10. Page 3, line 5: the word "suppression" here seems too strong – maybe "limitation". Unfortunately, the annual cycle of baroclinicity in the NH Pacific is not reported in any figure - if I am not wrong - in this paper.
11. Page 3, line 15: self-sustain – why "self"?
12. Page 4, line 11: "isobaric levels" in place of "vertical levels" (also at line 4 of page 5).
13. Page 4, lines 18-19: the sentence "For the pressure analysis 8 pressure levels..." should be dropped because it is a repetition.
14. Page 4, lines 27-28 ("INM-CM3..."): also this sentence is redundant.
15. Page 5, lines 21 and following: for the sake of clarity, pls. introduce here the full definition, specifying averages (zonal, vertical, latitudinal) of both quantities – part of the specifications is given below (lines 29 and following), but it should be anticipated before any further description and comment for readability. U is the zonal wind component, not the horizontal wind component.
16. Page 5, lines 29: "is averaged over" in place of "takes into account" (too generic).

17. Page 7, line 22 (and elsewhere): the standard acronym for geopotential height is GPH, not GPT.
18. Page 7, line 22: "... variance in the *same* midlatitude belt". Moreover, the word "variance" is not used here in its technical sense and should be better defined here or substituted with "variability" (also at line 25 and maybe elsewhere).
19. Line 24: is the 15-day moving average "tapered"? If not, some high frequency noise is retained (but this problem may affect the high frequency part of the spectra, not the 6-month period).
20. Page 8, lines 1-3: perhaps insert here (or also in the conclusions) a comment referring to the results of the cited paper of Donohoe and Battisti.
21. Page 8, line 4: pls. specify better which aspects of the MSLP cycles can be related to SAO.
22. Page 8, lines 16, 17 (and elsewhere, as for examples lines 31 and 32): I think the article "the" should be put in front of NH and SH (unless, of course, they are used as adjectives) – please try to be consistent throughout the text.
23. Page 8, line 18: perhaps "For a comparison", in place of "As an example".
24. Page 9, lines 1-2: the relationship between surface temperature and baroclinicity cycles cannot be deduced by the simple analysis of this paper, considering that (in particular for the annual cycle) both quantities are subject to solar "forcing" (see also my general comments 1 and 3).
25. page 9, lines 9-10: ".... for the analysis ERA (and in Figures 8-9 for model output, described below), respectively".
26. Page 9, line 15: the word "correlation" should be intended here in the statistical sense, with no physical/dynamical implication (see comment 24 above).
27. Page 9, lines 33-34: again, a physical interpretation is missing here. The sentence "... the role of the semiannual variability in shaping eddy activity" is meaningless: "variability" is a physical/statistical property, not a physical factor.

Figures:

Figure 1 perhaps contains too many lines, with no labels - so it is not easy to distinguished the curves on the basis of colours only. I suggest to drop the GPH at 500 or 300 hPa, since they are very similar.

Figure 6: captions need to be corrected, because, unlike Fig. 5, this figure depicts spectra only for T2m, not for baroclinicity.

---

## Referee Comment (RC3) · Anonymous Referee #3 · 10 Aug 2016

The paper is generally well written and with a clear strategy. It contains some potentially interesting outcomes on the skills of global datasets (Renalysis and AOGCMs as well) in reproducing the most relevant harmonics (annual/semiannual) of some parameters of interest for mid-latitude synoptic variabiilty such as 2 meter temperature and mean baroclinicity (mostly related to the Available Potential Energy ). However, I think that some points could be better addressed before considering it for a final publication.

1) In several parts of the paper more explanations , comments and interpretations are needed. There are too many vague statements that should be better assessed and figures not adequately commented. In the specific comments below some examples are reported

[Figure]

2) In the context of the main objectives of this paper I have found the jump between the reanalysis and AOGCMs too large. One possible question is if the discrepancies observed in the representation of the first harmonics in the climate models are due to a lack in the description of main atmospheric processes or, alternatively, to the coupling with oceanic components. So, I strongly suggest to add to the present analysis some AMIP runs forced by observed SST. One possible candidates could be the AMIP runs recently produced in the ERA-CLIM project (Hersbach et al. 2015). Moreover, in order to have a longer reference dataset, the authors could also analyse the centennial re-analysis ERA-20C (Poli et al 2015) Hersbach, H., Peubey, C., Simmons, A., Berrisford, P., Poli, P. and Dee, D. (2015), ERA-20CM: a twentieth-century atmospheric model en-semble. Q.J.R. Meteorol. Soc.. doi: 10.1002/qj.2528 Poli P, Hersbach H, Berrisford P, Dee D, Simmons A. and Laloyaux P. (2015) ERA-20C Deterministic. ECMWF ERA Report Series 20, ECMWF, Shinfield Park, Reading

3) The quality of the figures should be improved to add readability to this work (see specific comments below).

Specific comments: Sec.2.1 As far as I know, in a recent paper Di Biagio et al (2014) have applied the same metrics introduced in Lucarini et al 2007 (here cited) on CMIP5 models. I suggest that this work should be here considered and the main outcomes should be taken into account, especially concerning the CMIP5 models analysed. Di Biagio, V., S. Calmanti, A. Dell'Aquila, and P. M. Ruti (2014), Northern Hemisphere winter midlatitude atmospheric variability in CMIP5 models, Geophys. Res. Lett., 41, doi:10.1002/2013GL058928.

Sec 3.1 In fig.1 a legend for the different lines should be added. Moreover, additional explanations (or even just references) about the relevant features of SH (for instance the october relative minimum in the meridional geopotential gradient) are here required I have found fig.2-3 almost useless. I suggest to remove them or alternatively to merge fig. 2 and fig 3, as already done in Fig.4, to better highlight the differences in phase between T2m and the index of baroclinicity. However, the periodic features have been

already highlighted in Fig.1, so the authors should better explain the motivation of those figures, if they want to keep them further Similarly, I suggest to modify Fig.4 reporting for each model only the standardised mean seasonal cycle of the two indicators (as in Fig.1). Also some additional comments about the different skills of the models are here needed.

Sec. 3.2 Why the authors have chosen the integer values p=3 and K=5 in the MTM method, and 7 degrees of freedom for the Chi-squared distribution? Could the authors add some details and explanations on that point? Fig-5-9: To improve the readability of the figures I suggest to report in x-axis the period (in months) instead of frequency Fig. 6-9 Please add the considered variable in the title of each figures The spectral analysis for the CMIP3/5 models is applied to 100 years instead of 36 years for Era-Interim. To be consistent, records with the same length should be considered. I suggest to re-run this analysis by considering records with comparable sizes. To have a longer record for the reanalysis, I would suggest, as already mentioned, the adoption of ERA-20C centennial reanalysis In this section, an additional analysis considering AMIP runs could be quite appealing to check if the discrepancies here arisen in CMIP3/5 models are due to the use of coupled models.

Sec.4 " Present findings contribute to better characterize the cyclic response of current global atmosphere-ocean models to the external solar forcing that is of particular interest for seasonal forecasts". Here some additional comments about seasonal forecast, or even just some references, are required
* * *

---

## Author Comment (AC1) · 22 Sep 2016

*Reply to the Reviews on*

**Annual and semiannual cycles of midlatitude surface temperature and baroclinicity: reanalysis data and AOGCMs simulations**

by
Valerio Lembo, Isabella Bordi, Antonio Speranza

We thank the Reviewers for their comments and suggestions. Below we quote each comment and provide our brief response to it.

**Reviewer #1**
The authors investigate the annual and semiannual cycles of atmospheric near surface temperature and baroclinicity (maximum Eady growth rate) in midlatitudes. They analyze the statistical relationship between the two quantities, and assess the ability of CMIP3 and CEMIP5 models to reproduce properties derived from ERA Interim reanalysis. The results show high coherence between the two variables for both the annual and the semiannual cycle, but with different relative phases. The CMIP models show good agreement with reanalysis for coherence at annual and semiannual frequency. For relative phase at semiannual frequency larger differences between models and reanalysis and among the models are observed. Improvements for CMIP5 models compared to CMIP3 are found.

General To test the ability of climate models to simulate the present day climate, and, thus, to give some confidence in their projection of potential future climates the simulated annual and semiannual cycles are appropriate testbeds. In addition, near surface temperature and baroclinicity (or maximum Eady growth rate, as an indicator for eddy activity) are important quantities defining the climate state. Thus, the study conducted here addresses relevant scientific questions and fits the scope of Earth System Dynamics. The methodology applied is sound and overall presentation is well structured and clear. Though the paper does not present novel concepts, ideas, tools or data, I think that it presents potentially valuable new results. However, I have three specific points the authors need to address before I can recommend the paper to be accepted.

Specific 1) I appreciate the spectral analysis of near surface temperature and baroclinicity individually, but the significance of doing the spectral coherence/phase analysis between both as presented in this paper is not clear to me. The authors only give vague motivation for doing the coherence analysis by saying that '... temperature is taken as a proxy in climate change studies' (P2L18) and '... investigating the possible relationship...may help to better understand the role played by the latter...' (P3L16+17). For me, the combined analysis of near surface temperature and baroclinicity appears a bit arbitrary and the meaning/interpretation of the results seems little conclusive:
(i) The results seem to indicate that the coherence is mainly restricted to the annual cycle (and higher harmonics) which may simply because both variables have an annual cycle without any physical relation between them. At the moment, I cannot see a significant indication for a relation other than a pure statistical one (see also 2). If understanding of the relation between near surface temperature and baroclinicity is an aim of this study, there need to be some more statements/discussions on this in the conclusion. At the moment the paper gives some discussions about baroclinicity and temperature gradients (e.g. SAO), but the link to temperature itself is missing, or appears to be an indirect one which is related to changes in near surface temperature gradients. In other words: for analyzing a possible relation between near surface temperature and

baroclinicity the (equator-to-pole) temperature gradient may be a more obvious quantity. Why do the authors not consider this directly?

**Reply:** As known, the temperature cycle on the annual time scale responds directly to the solar irradiance changes and feedbacks between the different components of the Earth's system, like the sea-ice-albedo feedback, can amplify or dampen the response.

On the one hand, the annual cycle of the global mean net radiation at the top of the atmosphere (defined as the difference between the downward absorbed solar radiation and the outgoing longwave radiation, $R_{TOA} = R_{SR} - R_{LW}$) is of the order of 20 W/m$^2$ or, integrated globally, about 10 PW (e.g., Fasullo and Trenberth 2008). On the other hand, the global climate forcing (i.e., the change of the planetary energy balance) due to the total greenhouse gases is estimated to be of less magnitude, about 3 W/m$^2$ or 1.5 PW, based on the change in gas concentrations since 1750 (NOAA 2016). Thus, the annual/semiannual cycles are the leading natural changes which the atmosphere experience every year, and their proper representation can be considered the starting point for any climate projection. This applies also in relation to GCM climate simulations that have highlighted changes in the seasonality of surface temperature (amplitude and phase) in response to the increasing greenhouse gases concentration (e.g., Dwyer et al. 2012 and references therein).

Furthermore, one of the topical scientific open questions is the link between climate warming and weather variability, with particular reference to changes in the intensity and frequency of extratropical cyclones/anticyclones. Such changes are thought to be closely related to a reduction or intensification of the baroclinicity of the lower troposphere under warming climate conditions (e.g., McCabe et al. 2001). Also, comprehensive GCM studies suggest that the observed widening of the Hadley circulation may be due to the poleward shift in baroclinic eddy activity as a consequence of global warming, particularly an increase of global mean temperature (e.g., Frierson et al. 2007; Levine and Schneider 2015).

On these grounds, it appears of interest to investigate whether coupled models show any (amplitude or phase) direct relationship between zonal mean near surface temperature and baroclinicity. As known, baroclinic activity has its source in the available potential energy in the form of meridional temperature gradients, which in turn are affected by the eddies heat transports themselves. In the present paper, we do not address this specific aspect since it would require a comprehensive analysis of the heat fluxes induced by the eddies in the meridional plane.

Furthermore, we notice that the baroclinicity index here considered, which is proportional to the tropospheric meridional temperature gradient, is independent from the zonal mean near surface temperature; it would be not properly the same if the meridional gradient of T2m would be considered in place of T2m, due to the expected degree of dependence between tropospheric and near surface temperature gradients.

Lastly, it is worth noticing that high coherence values are found both for the annual and semiannual frequencies (see Figures 7–9) and are not restricted to the annual cycle.

Additional references are:

Dwyer, J.G., Biasutti, M., Sobel, A.H.: Projected changes in the seasonal cycle of surface temperature, J. Climate, 25, 6359–6374, 2012.

Fasullo, J.T, and Trenberth, K.E.: The annual cycle of the energy budget. Part I: General mean and land-ocean exchanges, J. Climate, 21, 2297–2312, 2008.

Frierson, D.M.W., Lu, J., Chen, G.: Width of the Hadley cell in simple and comprehensive general circulation models, Geophys. Res. Lett., 34, L18804, doi:10.1029/2007GL031115, 2007.

Levine, X. J., and Schneider, T.: Baroclinic eddies and the extent of the Hadley circulation: an idealized GCM study, J. Atmos. Sci., 72, 2744–2761, 2015.

McCabe, G.J., Clark, M.P., Serreze, M.C.: Trends in Northern Hemisphere surface cyclone frequency and intensity, J. Climate, 14:2763−2768, 2001.

NOAA (National Oceanic and Atmospheric Administration). 2016. The NOAA Annual Greenhouse Gas Index. Accessed June 2016. www.esrl.noaa.gov/gmd/aggi.

(ii) A second aim of the study is the evaluation of CMIP models. Given the somehow unclear (physical) interpretation of the observed coherences and relative phases between baroclinicity and temperature (see above), looking at the absolute phases of both variables may be of significant benefit. This may also help to identify the source of the discrepancies with respect to the semiannual cycle: do the phases of the baroclinicity or the near surface temperature (or both) differ? In this respect: L12 of the abstract suggests that also the absolute phases are considered.

**Reply:** Absolute phases of observed (ERAI) and modeled (AOGCMs) baroclinicity index, and of 2-meter temperature are listed below in Table 1.
Almost all phases are lagged less than about 1 month with respect to ERAI (i.e., about $30^o$ for the annual phase and $60^o$ for the semiannual one) and better agreement is found for CMIP5. Being 1 month the sampling time of the time series, results suggest that models are almost able to reproduce the phases of the reanalysis data. However, the observed coherences and relative phases between baroclinicity and 2-meter temperature may be partially affected by such not a perfect in phase relationship between model data and reanalysis. This aspect should be taken into account as a possible source of uncertainty when results of bivariate phase spectra analysis are compared and interpreted.

Table 1. Absolute phases of baroclinicity index and 2-meter temperature for the reanalysis ERAI, models CMIP3 and CMIP5.

| | $T2m$ | | | | $\sigma_{AB}$ | | | |
|---|---|---|---|---|---|---|---|---|
| | Annual (NH) | Semiannual (NH) | Annual (SH) | Semiannual (SH) | Annual (NH) | Semiannual (NH) | Annual (SH) | Semiannual (SH) |
| **ERAI** | 148.68 | 273.59 | 335.56 | 343.02 | 2.88 | 172.22 | 190.00 | 217.6 |
| **CMIP3** | | | | | | | | |
| CGCM3.1 | 139.25 | 216.24 | 330.89 | 78.37 | 3.11 | 183.29 | 187.13 | 217.6 |
| ECHAM5 | 143.70 | 264.36 | 326.01 | 327.07 | 6.32 | 178.85 | 192.6 | 213.5 |
| FGOALS-g1.0 | 144.00 | 193.64 | 335.70 | 276 | 359.5 | 175.46 | 197.41 | 197.66 |
| GFDL-CM2.1 | 153.14 | 282.19 | 331.17 | 320.40 | 10.82 | 188.20 | 163.28 | 202.22 |
| INM-CM3.0 | 151.25 | 259.11 | 338.5 | 311.3 | 3.27 | 198.7 | 196.5 | 219.13 |
| MIROC3.2 | 146.03 | 283.37 | 328.62 | 315.9 | 0.92 | 179.81 | 223.40 | 239.80 |
| **CMIP5** | | | | | | | | |
| CanCM4 | 145.74 | 290.01 | 330.59 | 14.35 | 1.98 | 175.38 | 180.66 | 215.62 |
| FGOALS-g2 | 144.02 | 262.26 | 335.17 | 318.31 | 8.84 | 158.86 | 189.91 | 202.38 |
| GFDL-CM3 | 123.36 | 228.3 | 331.79 | 316.85 | 12.65 | 176.35 | 204.01 | 205.80 |
| INM-CM4 | 150.59 | 294.39 | 332.38 | 1.31 | 6.03 | 204.06 | 197.53 | 222.25 |
| MIROC5 | 140.95 | 261.60 | 325.38 | 310.26 | 358.87 | 178.02 | 182.64 | 208.35 |
| MPI-ESM-MR | 143.44 | 263.32 | 326.26 | 319.93 | 9.98 | 188.51 | 191.75 | 223.12 |

2) P12L11-13: The authors state that '... the presence of a statistically significant semiannual peak in surface temperature spectral estimates, may suggest that the internal forcing exerted by baroclinic eddies play a role in modulating the annual cycle'. I do not understand this statement: Why does a significant semiannual peak observed in the surface temperature indicate a forcing different from the solar one, and, moreover, a forcing related to baroclinic eddies? Please clarify.

**Reply:** The external solar forcing is directly responsible for the annual periodicity. The existence of the semiannual period in both near surface temperature and baroclinicity index might be related to the result of a feedback mechanism between baroclinic activity and near surface temperature through the effects of the eddies heat transports (i.e., their impact on the meridional temperature gradients), in analogy with what happens in SAO phenomenon.

3) In the paper the authors use surface temperature and near surface temperature as synonymously, but actually atmospheric near surface temperature is used. Thus, to be more precise/clear, the authors may use the term near surface temperature throughout the paper. I do not think that the results are very sensitive to the particular choice of surface or near surface temperature. However, there are major differences between both variables, e.g. while the surface temperature is prognostic in the models, the near surface temperature is diagnosed from the lower atmospheric levels by using a somehow artificial role related to the computation of the surface fluxes.

**Reply:** For the analysis we use the 2-meter air temperature as specified in section 2.1. We revise the text accordingly to avoid any misunderstanding.

4) The summary and conclusions section suggests that zonally (and vertically) averaged values are used in this study (P11L7). But, from the results section we learn that for the northern hemisphere the analysis focus on the Pacific. This should be made clearer (more consistent) in the summary and conclusions (Otherwise, e.g., the origins of results (iii) and (v) are unclear for readers focusing on the summary and conclusions only).

**Reply:** The Reviewer is right since in the concluding section we have not specified that after a preliminary analysis of the NH zonally averaged fields, we focus on the NH Pacific sector, where a more pronounced and statistically significant peak at the semiannual frequency is found for 2-meter temperature. We revise the text accordingly.

**Reviewer #2**
This paper analyses some statistical properties of the global atmosphere in comparison with results obtained from general circulation/climate models. In particular, the annual and semiannual cycles of near-surface temperature and of an index of baroclinic activity are examined in detail.
The technical execution is basically correct and applied analysis methods are reasonably funded and appropriate. At least part of the results is original and interesting. What is basically missing, in my opinion, is a deeper physical discussion of the working hypotheses and of the results. This deficiency makes this paper to appear more as a preliminary contribution than a conclusive one (the authors are aware of that, as they write in several circumstances "... require further investigation" or so). I think that this paper would gain in quality if a discussion is provided at some appropriate points and in the conclusions, as expressed in the following general comments. Therefore, I consider that this paper can be made worth of publication if a revision will be made in response to my indications given below.
General comments.
    1. Two parameters are chosen for the analysis, the near-surface temperature (T2m) and a

tropospheric-averaged baroclinicity index (sBI), expressing a theoretical grow rate of baroclinic instability. I think that this choice of the parameters should be better explained. In fact, the physical link between such quantities is very complex and can be very different depending on the time scale of the processes involved. The fact that they are strictly correlated (actually, anti-correlated) in the annual cycle is quite obvious, with reference, for example, to the results exposed in the paper by Donohoe and Battisti (2003, cited in this paper): considering the large impact on the atmospheric seasonal heating due to direct absorption of the solar radiation (not neglecting, of course, the role of land and ocean fluxes), and considering the very large annual cycle of both solar radiation and of the meridional gradient of it (which even changes it sign between solstices), it is not surprising that both quantities are strongly modulated by the astronomical cycle. However, this does not mean that the two quantities have a simple relation between each other, stemming from the atmospheric circulation dynamics! It seems that the choice of T2m has been made having in mind the application of climatic models to the long-term global changes (see for example line 18 of page 2), but this is something different form considering the seasonal-annual variability. On the other hand, the baroclinic index may be a subtler indicator of long term variability, while it is closely related to many aspects of the general circulation dynamics, including of course mid-latitude baroclinic "turbulence" and its limiting factor.

**Reply:** The motivation of the present study is twofold: (i) to assess the ability of AGCMs to correctly simulate the annual and semiannual cycles in near-surface temperature and tropospheric baroclinicity; (ii) to investigate if there is some amplitude/phase relationship between the two variables. The annual periodicity is the main cycle of variability of many atmospheric variables since it is strictly related to the annual solar irradiance variation. Such cycle is by far larger compared with the estimated global climate forcing, and hence models should be able to well reproduce such a periodicity. Furthermore, the choice of the variables is related to the general problem of climate warming that several studies have identified as the cause of a change/shift of baroclinic activity at midlatitudes. See also the reply to the Reviewer #1 (point 1).

2. Concerning the evaluation of the AOGCMs, the intercomparison based on seasonal-annual harmonics may not be very appropriate to assess model performance with respect to long-time variability, associated for example with the dynamics of slow processes like those related to ocean deep circulation, evolution of the cryosphere and the biosphere etc. However, I agree that a "good" climatic model should behave well also in simulating/predicting intra-annual time scales.

**Reply:** As discussed in the reply to the Reviewer #1 (point 1), the annual cycle of the global mean net radiation at the top of the atmosphere is of the order of 20 $W/m^2$, which is much greater than the global climate forcing due to the total greenhouse gases estimated to be about 3 $W/m^2$. Thus, a reliable representation of the annual/semiannual variability by AOGCMs is mandatory to be confident on future climate simulations. The present study wishes to focus on this issue providing some preliminary investigations. Additional efforts should be done to check the sensitivity of the present results to changes in model parameters/schemes, like the different representations of the ocean dynamics or biosphere. However, this remains beyond the scope of the paper because a comprehensive analysis with ad hoc simulations is required.

3. Regarding the appreciation of annual vs semiannual cycles, one should consider, in principles, that while the astronomical forcing is basically annual ta mid latitudes, it is basically semiannual at the equator. The analysis of the paper is restricted to the mid latitudes, but how does the intertropical semiannual variability affect the extratropical variability? The authors correctly mention possible relations with the SAO, but a more specific discussion (at least more detailed references to the literature) is needed in this respect, regarding SAO and perhaps more general semiannual aspects of the atmospheric

global circulation.

**Reply:** The aim of the present study is to investigate whether modern AOGCMs properly represent the annual/semiannual cycles of near surface temperature and baroclinicity at midlatitudes since one of the topical question is the relationship between changes in surface temperature and extreme weather events. The study of the possible teleconnection between the intertropical seasonal variability and the midlatitude circulation is of interest but would depart from the main target of the paper. In the revised text (concluding section) we mention about this problem and include some references (e.g., Vimont et al. 2001, 2003).

Vimont, D.J., Battisti, D.S., Hirst, A.C.: Footprinting: A seasonal connection between the tropics and midlatitudes, Geophys. Res. Lett., 28, 3923–3926, 2001.

Vimont, D.J., Wallace, J.M., Battisti, D.S.: The seasonal footprinting mechanism in the Pacific: implications for ENSO, J. Climate, 16, 2668–2675, 2003.

4. As anticipated above, the discussion of the results and the conclusion miss a sufficient consideration of the physical implications in terms of known properties of the general circulation of the atmosphere. I am asking for at least a somehow better indication of the perspectives derived from the results of this paper in terms of (a) possible relationships between the ERAI-based results and some known aspects of the global atmospheric dynamics (for example the SAO), and (b) somehow less vague indications of the possible implications concerning AOGCMs' performance, so as to provide modellers with more specific hints (only the horizontal resolution is mentioned at line 33 of page 10, which is perhaps not the most important aspect: I assume that many more subtle physical and dynamical problems have still to be solved to improve the model accuracy).

**Reply:** We think that a comprehensive evaluation of AOGCMs' performance in terms of capability to reproduce the observed annual and semiannual cycles should be the natural extension of the present preliminary study. At the stage of the present analysis, it is not possible to identify specific indications for modellers to improve the model accuracy or major details concerning selected phenomena like SAO. Comparisons with additional reanalysis data or observations, as well as sensitivity studies (for example those concerning model resolution or parameterization schemes) should be carried out; for a given aspect to be analysed a set of model experiments must be carried out and inter- and intra-model comparisons should be taken into account.

Minor comments.
1. Title (and also in the text: for example, at line 9 page 1, lines 21 and 24 page 2): it is not clear (before reading the definition in Sect. 2.2) if the word "surface" is referred also to "baroclinicity". So I suggest to use "tropospheric baroclinicity" or something similar in the tile and in the text, before the exact definition is given in Sect. 2.

**Reply:** To avoid any confusion we changed the title as suggested: *"Annual and semiannual cycles of midlatitude surface temperature and tropospheric baroclinicity: reanalysis data and AOGCMs simulations"*.

2. Page 2, line 2: contribute.

**Reply:** We agree, we have corrected the misprint.

3. Pls. specify that "seasonal heating" is intended as the heating variability after subtracting the annual mean.

**Reply:** We revised the text as suggested: *"seasonal heating (i.e., the heating variability after subtracting the annual mean)"*.

4. Page 2, line 8: "... and the atmosphere is heated from below": this seems to contradict the previous sentence that most of the (seasonal) atmospheric heating is due to direct atmospheric absorption. Pls. clarify.

**Reply:** It is not a contradiction since the dominant oscillation is the annual one that is strictly related to the insolation. As known, the atmosphere is almost transparent to the short ware radiation that is absorbed by the Earth's surface.

5. Page 2, line 17: perhaps "... the first efforts *in understanding* the climate impact...".
**Reply:** We agree, we changed the text accordingly.

6. Page 2, line 18: proxy of/for what? of globally averaged temperature? Pls. specify.
**Reply:** Yes, of globally averaged temperature, we specified in the revised text.

7. Page 2, line 19: "*its* key role".
**Reply:** We changed as suggested.

8. Page 2, line 24: AOGCM in place of GCM (as elsewhere in the text).
**Reply:** We changed as suggested.

9. Page 3, lines 2-3: consider this change "there are regions of enhanced eddy activity ("storm tracks"), where... decay.".
**Reply:** We changed as suggested.

10. Page 3, line 5: the word "suppression" here seems too strong – maybe "limitation". Unfortunately, the annual cycle of baroclinicity in the NH Pacific is not reported in any figure - if I am not wrong - in this paper.
**Reply:** The word "suppression" is here used as in previous papers documenting the seasonal behaviour of baroclinic activity in NH Pacific (see for example Nakamura 1992). For this reason we think that it is better to retain the word "suppression" to identify the phenomenon in question.
The Reviewer is right we did not show the time series of the baroclinicity index in NH Pacific because it does not add any additional information to the analysis; it is worth noticing that the problem of the statistical significance at the semiannual frequency in the NH involves only near surface temperature.

11. Page 3, line 15: self-sustain – why "self"?
**Reply:** Perhaps "self " is not appropriate to explain the concept of continuity in time. We use "*to continuously sustain*".

12. Page 4, line 11: "isobaric levels" in place of "vertical levels" (also at line 4 of page 5).
**Reply:** We changed the text as suggested.

13. Page 4, lines 18-19: the sentence "For the pressure analysis 8 pressure levels..." should be dropped because it is a repetition.
**Reply:** We agree.

14. Page 4, lines 27-28 ("INM-CM3..."): also this sentence is redundant.
**Reply:** We agree.

15. Page 5, lines 21 and following: for the sake of clarity, pls. introduce here the full definition, specifying averages (zonal, vertical, latitudinal) of both quantities – part of the specifications is given below (lines 29 and following), but it should be anticipated

before any further description and comment for readability. U is the zonal wind component, not the horizontal wind component.

**Reply:** We agree, we revised the text accordingly.

16. Page 5, lines 29: "is averaged over" in place of "takes into account" (too generic).
**Reply:** We agree.

17. Page 7, line 22 (and elsewhere): the standard acronym for geopotential height is GPH, not GPT.
**Reply:** We changed in the text and figures.

18. Page 7, line 22: "... variance in the *same* midlatitude belt". Moreover, the word "variance" is not used here in its technical sense and should be better defined here or substituted with "variability" (also at line 25 and maybe elsewhere).
**Reply:** Technically, it is the variance of the mean sea level pressure.

19. Line 24: is the 15-day moving average "tapered"? If not, some high frequency noise is retained (but this problem may affect the high frequency part of the spectra, not the 6-month period).
**Reply:** Yes, we apply the 15-day moving average (tapered) just to filter out high frequencies, although we agree that it is not necessary for the objective of the paper; it is here applied just for illustrative purpose.

20. Page 8, lines 1-3: perhaps insert here (or also in the conclusions) a comment referring to the results of the cited paper of Donohoe and Battisti.
**Reply:** We revise the text accordingly: "*It is worth noticing that results are in agreement with those obtained by Donohoe and Battisti (2013) showing that atmospheric temperature lags the insolation by approximately 30 days in the northern and 40 days in the southern extratropics, respectively.*"

21. Page 8, line 4: pls. specify better which aspects of the MSLP cycles can be related to SAO.
**Reply:** As documented by Meehl (1991), the SAO phenomenon is evident in monthly mean maps of observed mean sea level pressure (SLP). The trough of SLP minimum is farthest south and deepest in March and September, while it is farthest north and weakest in June and December (their Figure 1). Such a movement of the circumpolar trough is associated with changes in the cyclone activity in extensive areas. However, the SAO phenomenon is also evident throughout the depth of the troposphere, for example in 500 mb temperature.

22. Page 8, lines 16, 17 (and elsewhere, as for examples lines 31 and 32): I think the article "the" should be put in front of NH and SH (unless, of course, they are used as adjectives) – please try to be consistent throughout the text.
**Reply:** We changed as suggested throughout the text.

23. Page 8, line 18: perhaps "For a comparison", in place of "As an example".
**Reply:** We changed in the text.

24. Page 9, lines 1-2: the relationship between surface temperature and baroclinicity cycles cannot be deduced by the simple analysis of this paper, considering that (in particular for the annual cycle) both quantities are subject to solar "forcing" (see also my general comments 1 and 3).
**Reply:** Here we intend to point out that in analysing the relative phase and coherence between T2m

and $\sigma_{AB}$ at the annual and semiannual components, the statistically significance at both frequencies is required. The NH Pacific region is selected for this reason.

> 25. page 9, lines 9-10: ".... for the analysis ERA (and in Figures 8-9 for model output, described below), respectively".

**Reply:** We changed in the text.

> 26. Page 9, line 15: the word "correlation" should be intended here in the statistical sense, with no physical/dynamical implication (see comment 24 above).

**Reply:** We agree. However, by definition, the coherence spectrum is analogous to the conventional correlation coefficient.

> 27. Page 9, lines 33-34: again, a physical interpretation is missing here. The sentence "... the role of the semiannual variability in shaping eddy activity" is meaningless: "variability" is a physical/statistical property, not a physical factor.

**Reply:** We changed "shaping" with "*modulating*" that is more appropriate.

Figures:
Figure 1 perhaps contains too many lines, with no labels - so it is not easy to distinguished the curves on the basis of colours only. I suggest to drop the GPH at 500 or 300 hPa, since they are very similar.
**Reply:** We have revised the figure as follow.

[Figure]

[Figure]

Figure 6: captions need to be corrected, because, unlike Fig. 5, this figure depicts spectra only for T2m, not for baroclinicity.
**Reply:** We have corrected the caption.

**Reviewer #3**
The paper is generally well written and with a clear strategy. It contains some potentially interesting outcomes on the skills of global datasets (Renalysis and AOGCMs as well) in reproducing the most relevant harmonics (annual/semiannual) of some parameters of interest for mid-latitude synoptic variabiilty such as 2 meter temperature and mean baroclinicity (mostly related to the Available Potential Energy). However, I think that some points could be better addressed before considering it for a final publication.

1) In several parts of the paper more explanations, comments and interpretations are needed. There are too many vague statements that should be better assessed and figures not adequately commented. In the specific comments below some examples are reported

2) In the context of the main objectives of this paper I have found the jump between the reanalysis and AOGCMs too large. One possible question is if the discrepancies observed in the representation of the first harmonics in the climate models are due to a lack in the description of main atmospheric processes or, alternatively, to the coupling with oceanic components. So, I strongly suggest to add to the present analysis some AMIP runs forced by observed SST. One possible candidates could be the AMIP runs recently produced in the ERA-CLIM project (Hersbach et al. 2015). Moreover, in order to have a longer reference dataset, the authors could also analyse the centennial re- analysis ERA-20C (Poli et al 2015) Hersbach, H., Peubey, C., Simmons, A., Berrisford, P., Poli, P. and Dee, D. (2015), ERA-20CM: a twentieth-century atmospheric model ensemble. Q.J.R. Meteorol. Soc.. doi: 10.1002/qj.2528 Poli P, Hersbach H, Berrisford P, Dee D, Simmons A. and Laloyaux P. (2015) ERA-20C Deterministic. ECMWF ERA Report Series 20, ECMWF, Shinfield Park, Reading

**Reply:** As suggested by the Reviewer, we have considered six AMIP models (CanCM4, FGOALS-g2, GFDL-CM3, INM-CM4, MIROC5, MPI-ESM-MR) forced by observed SST and ERA-20CM developed by the ECMWF. For AMIP runs we have considered the common time section 1979–2009, while for ERA-20CM 1979–2011. Results are shown in the figure below (ERAI in green, AMIP in magenta, ERA-20CM in blue, CIMIP5 in red).

Results appear comparable with those obtained with CMIP5, with a general slight improvement at the annual frequency ($10^{o}$–$15^{o}$). Using AMIP, small improvements (about $15^{o}$) are obtained for the Pacific sector at the semiannual frequency, while the model INM-CM4 shows a smaller relative phase in SH midlatitudes when compared with other models. At the stage of the present analysis, results suggest that the impact of observed SST on the modelled relative phase is primarily on the annual cycle (though limited to a few degrees) and, as expected, on the NH Pacific ocean sector.

Results for ERA-20CM appear consistent with those for ERAI.

[Figure]

3) The quality of the figures should be improved to add readability to this work (see specific comments below).

Specific comments: Sec.2.1 As far as I know, in a recent paper Di Biagio et al (2014) have applied the same metrics introduced in Lucarini et al 2007 (here cited) on CMIP5 models. I suggest that this work should be here considered and the main outcomes should be taken into account, especially concerning the CMIP5 models analysed. Di Biagio, V., S. Calmanti, A. Dell'Aquila, and P. M. Ruti (2014), Northern Hemisphere winter midlatitude atmospheric variability in CMIP5 models, Geophys. Res. Lett., 41, doi:10.1002/2013GL058928.
**Reply:** In the revised version of the paper we introduce the paper by Di Biagio et al. (2014). In that paper the authors evaluated whether CMIP3 and CIMIP5 models predict future shifts in the global baroclinic eddies and planetary scale wave activity. Results suggested no significant improvements with CMIP5 ensemble and limited changes of baroclinic activity in RCP.4.5 scenarios. Differently, in the present analysis CMIP5 ensemble shows a better representation of the annual/semiannual cycles when compared with the previous version CMIP3.

Sec 3.1 In fig.1 a legend for the different lines should be added. Moreover, additional explanations (or even just references) about the relevant features of SH (for instance the october relative minimum in the meridional geopotential gradient) are here required I have found fig.2-3 almost useless. I suggest to remove them or alternatively to merge fig. 2 and fig 3, as already done in Fig.4, to better highlight the differences in phase between T2m and the index of baroclinicity. However, the periodic features have been already highlighted in Fig.1, so the authors should better explain the motivation of those figures, if they want to keep them further Similarly, I suggest to modify Fig.4 reporting for each model only the standardised mean seasonal cycle of the two indicators (as in Fig.1).
Also some additional comments about the different skills of the models are here needed.
**Reply:** We don't agree that Figure 2–4 are almost useless. We firmly believe that showing data that are subjected to the analysis should be the starting point of any scientific investigation: it helps in illustrating the phenomenon under investigation, provides a quantification of the variables in question (for this reason we have not standardized the reanalysis time series), and may suggest what can be expected just from a visual inspection. In the present case, for

example, time series show the different amplitudes of the considered variables in the two hemispheres, display evidences of the annual and semiannual cycles in baroclinicity index and only of the annual cycle in near surface temperature.

Sec. 3.2 Why the authors have chosen the integer values p=3 and K=5 in the MTM method, and 7 degrees of freedom for the Chi-squared distribution? Could the authors add some details and explanations on that point?
Fig-5-9: To improve the readability of the figures I suggest to report in x-axis the period (in months) instead of frequency
Fig. 6-9 Please add the considered variable in the title of each figures
The spectral analysis for the CMIP3/5 models is applied to 100 years instead of 36 years for Era-Interim. To be consistent, records with the same length should be considered. I suggest to re-run this analysis by considering records with comparable sizes. To have a longer record for the reanalysis, I would suggest, as already mentioned, the adoption of ERA- 20C centennial reanalysis In this section, an additional analysis considering AMIP runs could be quite appealing to check if the discrepancies here arisen in CMIP3/5 models are due to the use of coupled models.
**Reply:** For the choice of MTM parameters we followed Ghil et al. (2002) quoted in the paper; it represents a reasonable compromise between bias and variance of the estimator, and depends on the record length. In our case, after some trials we selected K = 5 and p = (K + 1)/2.
In the new Figure 6 we have added the title with the name of the variable considered.
Usually power spectra are shown in the frequency domain. In our case the annual and semiannual frequencies are easily identified by 1 and 2 year$^{-1}$; thus, we think that expressing the period in months in place of frequency in year$^{-1}$ does not add any useful information.
We have repeated the analysis by considering common/comparable record length (i.e., 32 years). Results are shown in the following figures for ERAI (green), CMIP3 (blue) and CMIP5 (red). Error bars of the original analysis are in solid lines, while those for the common record length in dashed lines. As expected (time series clearly show the annual periodicity), results show a good agreement with not substantial discrepancies.
For AMIP and ERA-20CM results see the reply to point 2) above.

[Figure]

[Figure]

Sec.4 " Present findings contribute to better characterize the cyclic response of current global atmosphere-ocean models to the external solar forcing that is of particular interest for seasonal forecasts". Here some additional comments about seasonal forecast, or even just some references, are required.

**Reply:** We think that knowing whether the semiannual cycle characterizes or not the climate variability of a given midlatitude region may be useful to better verify (and eventually improve) the skill of seasonal forecasts. The annual and seasonal cycles are also important modulations of ENSO, which is certainly the dominant driver for seasonal prediction (e.g., Troccoli 2010).

Troccoli, A.: Seasonal climate forecasting, Meteorol. App., 17, 251–268, 2010.

---

## Author Comment (AC2) · 22 Sep 2016

The comment was uploaded in the form of a supplement:
http://www.earth-syst-dynam-discuss.net/esd-2016-28/esd-2016-28-AC2-supplement.pdf

---

## Author Comment (AC3) · 22 Sep 2016

The comment was uploaded in the form of a supplement:
http://www.earth-syst-dynam-discuss.net/esd-2016-28/esd-2016-28-AC3-
supplement.pdf
* * *

---

## Author Response (AR1)

Dear Editor,

we thank you for the prompt response. Below we comment the points that you have highlighted about the paper revision. In the revised text changes are in red.

Concerning the mentioned paper by Lucarini et al. (2014), the authors noticed that there is not a significant improvement in the CMIP5 multi-model mean average of the intensity and position of meridional heat transport peaks compared with CMIP3. Among the models that have been used, three of them (i.e., CGCM3.1/CanESM2[1], INMCM3/INMCM4, ECHAM5-MPIOM/MPI-ESM-MR)[2] have been also considered in our analysis.

ECHAM5-MPIOM and MPI-ESM-MR do not significantly differ in the intensities and positions of the peaks. The intensity of the peaks is the largest among the models considered by Lucarini et al. (2014). Changes are not relevant as well the annual and semi-annual relative phases. Annual phases are not consistent both in ECHAM5-MPIOM and MPI-ESM-MR. Moreover, both CGCM3.1/CanESM2 and INMCM3/INMCM4 exhibit significant changes in the intensities of the peaks in both hemispheres. Particularly, both CGCM3.1 and CanESM2 peaks are significantly shifted equatorwards in the SH. CanESM2 is the only relevant outlier in the CMIP5 multi-model ensemble considered by Lucarini et al. (2014). Concerning phases, INMCM family models improve their semiannual phases from version 3 to version 4. CGCM3.1 and CanCM4 incorrectly reproduce the SH semiannual phase. Particularly CanCM4 is the only model in our CMIP5 multi-model ensemble having an inconsistent SH semiannual phase.

Our results suggest a reduction in model discrepancies from CMIP3 to CMIP5 in the NH. This is not supported by evidences about the intensity of the atmospheric meridional heat transports, as suggested by Lucarini et al. (2014), although some improvements in terms of mutual agreement is found on the position of the SH peak in CMIP5 compared to CMIP3. Although meridional heat transports have not been explicitly considered in our analysis, results from Lucarini et al. (2014) have been compared with those from a blend of observations and reanalyses (Fasullo and Trenberth, 2008) in Lembo et al. (2016). There it is shown that models tend to underestimate intensities of total meridional heat transports in both hemispheres and atmospheric meridional heat transports in the NH. This provides some hints about the interpretation of results from the three pairs of individual models mentioned above. ECHAM5-MPIOM/MPI-ESM-MR pair is an outlier in terms of intensity of the peaks in both hemispheres, but it is closer to results from Fasullo and Trenberth (2008). Despite producing a correct meridional heat transport pattern, the relative annual phase is incorrectly reproduced in these models. Following arguments by Donohoe and Battisti (2012), about inconsistencies in the reproduction of the shortwave (SW) absorption as the main source of model discrepancies in the reproduction of meridional heat transports, we hypothesize that this inconsistencies might be reflected in an incorrect phasing between the astronomical forcing and the annual cycle of the mid-latitudinal temperature meridional gradient. The CGCM3.1/CanESM2 pair is affected by equatorward displacement of the SH peak position. It might be argued that this is linked to the incorrect representation of the SH semiannual relative phase. As mentioned in the manuscript, this results from the different phase of the near-surface temperature annual cycles at SH mid-latitudes and over Antarctica, leading to a semiannual behavior of meridional temperature gradients. The equatorward displacement of the peak might thus indicate a contraction of the Hadley circulation, with an equatorward displacement of the SH storm tracks, reducing the impact of higher latitudes annual cycle. As a consequence, this would lead to less consistency of the semiannual phase reproduction. These arguments were not the main focus of our analysis and it would certainly be of interest to investigate them in a successive work.

Concerning the non-linear response of the analyzed quantities to the periodic seasonal forcing, it is certainly interesting to take into account the Ruelle response theory (RRT), as in the suggested papers by Ragone et al. (2016) and Lucarini et al. (2016). The climate system is indeed proven to behave as an Axiom A dynamical system in a wide range of timescales, with the Chaotic Hypothesis holding,
* * *
[1] CanCM4 has actually been analyzed in our work. CanCM4 and CanESM2 only differ in the fact that the latter is also provided with an ocean carbon cycle module (CMOC1.2) and a land surface history (CTEM1). We believe that the comparison among the two model outputs is feasible, because long-term changes that are accounted for by CMOC1.2 and CTEM1 should not significantly impact the seasonality of the observed quantities.
[2] MIROC3.2 is mentioned in Table 1 but it is not shown in scatter plots of Figure 9b by Lucarini et al. (2014).

particularly when coarse-grained observables are considered (like the ones that we have considered). Nevertheless, one has to be careful when applying RRT for the study of the response to the seasonal forcing. The assumption of "weak perturbations of an autonomous unperturbed system" has to be assessed, depending on the considered observable. Whereas this assumption fits well when dealing with mid-latitude near-surface temperature, the magnitude of the seasonal perturbation to the baroclinic activity unperturbed state is such that, particularly in the NH, this is almost halved in summer compared to winter. Furthermore, it is non-trivial to perform a Schauder decomposition of the seasonal forcing, since the solar input equally evolves with time and changes its meridional pattern in the annual timescale. Non-linear processes are determinant for the response of the system in particular regions, as also evidenced by Lucarini et al. (2016). The sea-ice albedo feedback is of course important at high latitudes, whereas the meridional migration of the Hadley circulation, influences mid-latitudes meridionally displacing the poleward edges of the Hadely cells. Both processes act at very different scales, also including the annual time-scale, limiting the range of applicability of the response theory unless higher order Green functions are considered. From the point of view of the thermal inertia of the system, dealing with the seasonal cycle means considering a forcing that is far from being quasi-adiabatic (cfr. Fig. 4b in Lucarini et al., 2016).

Although beyond the scope of this work, it would certainly be interesting to investigate the range of applicability of the linear RRT for the prediction of the climate response to the seasonal solar forcing. One could hypothesize that in order to overcome previously mentioned limitations, holding the premises of building up the response function from the properties of the unperturbed state of the system, the fluctuation-dissipation theorem provides a feasible approach. The Ornstein-Uhlenbeck nature of the climate system in the seasonal timescale would then be necessarily assumed, i.e. the possibility to decompose the evolution of the system into a deterministic term and a state-independent Gaussian term. Recent works suggest that this approach is viable even when the seasonal cycle is retained (Gritsun and Branstator, 2016). Comparisons of the two approaches in the case of the seasonal forcing would be an exciting way to extend the study of climate sensitivities towards the high frequency edge of the power spectrum of climate variability.

[revised manuscript text omitted]

---

## Author Response (AR4)

*Reply to the Review on*

**Annual and semiannual cycles of midlatitude near-surface temperature and tropospheric baroclinicity: reanalysis data and AOGCMs simulations**

by
Valerio Lembo, Isabella Bordi, Antonio Speranza

We thank the Referee #2 for his/her comments. Changes are reported in *italic* and in the text in red.

**Reviewer #2**

I have appreciated the effort made by the authors in preparing the present revised version: the paper quality has improved. However, I do not think they have fully replied to my major comments; still, I am aware that some questions I posed were difficult to be answered within the limitations of this work. Also, perhaps I was not clear enough. The authors have also considered the majority of my minor comments. However, there are two of them which I think need additional consideration before I consider the paper worth of publication - I report here my original comment and the authors' reply:

4. Page 2, line 8: "... and the atmosphere is heated from below": this seems to contradict the previous sentence that most of the (seasonal) atmospheric heating is due to direct atmospheric absorption. Pls. clarify. Reply: It is not a contradiction since the dominant oscillation is the annual one that is strictly related to the insolation. As known, the atmosphere is almost transparent to the short ware radiation that is absorbed by the Earth's surface.
My comment: Donohoe and Battisti (2013) indicate that the "DIRECT" shortwave absorption" by the atmosphere is an important contribution for the annual cycle budget, so the heating from below is not the only relevant term. It seems to me that this means that the atmosphere CANNOT be considered almost transparent to shortwave radiation. I was myself a bit surprised by reading that, but I must confirm my previous comment: there is a contradiction in the sentence in the present paper and the authors should resolve it, after carefully reconsidering the results of D.&B (unless they have reasons to question them...).

**Reply:**
We agree with the reviewer that the two sentences in the first period of the paper might be misleading as such. Our aim was here to emphasize that there is a clear separation between the factors influencing the atmospheric heating at the seasonal and annual time scales. This is not meant to contradict Donohoe and Battisti (2013) by stating that the atmosphere is transparent to shortwave radiation, either in the seasonal, or in the annual timescales. Indeed Donohoe and Battisti (2013) clearly evidenced the dominant contribution of direct SW absorption for the seasonal cycle of the atmospheric heating.
In addition to that we might specify that in the annual mean roughly one fourth of incoming SW radiation is directly absorbed into the atmosphere (e.g. Wild et al., 2013, arguing that they amount to 340 and 79 W/m2, respectively). This estimate also accounts for a small part of SW absorption reflected by the surface, but this is marginal with respect to the SW radiation directly absorbed by the atmosphere. The heating from below is mainly effected by upward LW radiation, latent and sensible turbulent heat fluxes (accounting for 398, 84 and 20 $W/m^2$ respectively, according to Wild et al., 2013). To a zero-order approximation the net energy balance for a climate system in thermal equilibrium requires an annual mean LW emission to outer space of 240 $W/m^2$ (which equals the

amount of net SW radiation entering the system). Thus, the balance between LW radiation emitted by the surface and that exiting from the Top of the Atmosphere amounts to about 150 W/m$^2$, which in addition to the heat fluxes at the surface ensures that most of the atmospheric heating comes from the surface

We would appreciate if you might consider the revised version of the first part of the Introduction, where the annual-seasonal timescale separation is more clearly expressed:

*"The seasonal cycle of the heating of the atmosphere is one of the most prominent features of the Earth's climate (e.g., Kiehl and Trenberth, 1997; Trenberth and Stepaniak, 2004). A recent study by Donohoe and Battisti (2013) suggested that while in the annual average heating is dominated by upward energy fluxes from the surface, such as longwave, latent and sensible heat fluxes (e.g. Wild et al., 2013), most of the seasonal heating (i.e., the heating variability after subtracting the annual mean) is attributable to the direct shortwave absorption within the atmosphere, with an amplitude that is quite constant throughout the troposphere."*

Wild, M., Folini, D., Schär, C., Loed, N., Dutton, E.G., König-Langlo, G.: The global energy balance from a surface perspective, Climate Dyn., 40, 3107–3134, 2013.

27. Page 9, lines 33-34: again, a physical interpretation is missing here. The sentence "... the role of the semiannual variability in shaping eddy activity" is meaningless: "variability" is a physical/statistical property, not a physical factor. Reply: We changed "shaping" with "modulating" that is more appropriate.

My comment: "modulating" is better than "shaping" - however, it was the subject, not the verb, that I questioned. The "variability", without specifying of which physical quantity, cannot be considered to be a physical variable/factor... this is the point I tried to make. It is not only a matter of language. If not better specified, it risks to be a tautology: the modulation of eddy activity is itself a variability!

**Reply:**
We thank again the reviewer, because he/she evidenced that it was not clear enough in the text that we were resuming our results, referring to the modulation of the baroclinic eddy activity by means of the semiannual harmonic. This statement mainly involves the statistics of baroclinic eddies (whose 3–7 days timescale, often referred to as "synoptic timescale"); their activity is of course modulated by the incoming radiation annual cycle, affecting the annual cycle in the meridional temperature gradient, to whom the baroclinic index here used is proportional. Our results show that the baroclinic index is also characterized by a semiannual harmonic in both hemispheres, which modulates the synoptic scale baroclinic eddy activity. These results are in line with what was previously found in the SH mid-latitudes, and, at a regional level, in the NH Pacific mid-latitudes.

In order to improve the readability of this period, we agree that using the term "variability" is not appropriate, since it is not specified that we particularly refer to the six-month harmonic in the baroclinic index as a modulator of the synoptic scale baroclinic eddy activity. We thus changed the text in section 3.2, page 10 as following:

*"At the semiannual frequency, a phase shift of about 50° is observed in the SH and about 80° in the NH Pacific, with surface temperature delaying by about 1 month or more compared to the opposition of phase: results seem in agreement with the SAO phenomenon and may be indicative of the role of the semiannual harmonic in modulating NH synoptic time-scale baroclinic eddy activity (an example is the midwinter suppression characterizing the North Pacific storm tracks)."*